# Recent Developments in Warm Inflation

**Vahid Kamali** [1,2,3], **Meysam Motaharfar** [4] and **Rudnei O. Ramos** [3,5,*]

1  Department of Physics, Bu-Ali Sina (Aviccena) University, Hamedan 65178-016016, Iran
2  School of Physics, Institute for Research in Fundamental Sciences (IPM), Tehran 19538-33511, Iran
3  Physics Department, McGill University, Montreal, QC H3A 2T8, Canada
4  Department of Physics and Astronomy, Louisiana State University, Baton Rouge, LA 70803, USA
5  Departamento de Fisica Teorica, Universidade do Estado do Rio de Janeiro,
   Rio de Janeiro 20550-013, RJ, Brazil
*  Correspondence: rudnei@uerj.br

**Abstract:** Warm inflation, its different particle physics model implementations, and the implications of dissipative particle production for its cosmology are reviewed. First, we briefly present the background dynamics of warm inflation and contrast it with the cold inflation picture. An exposition of the space of parameters for different well-motivated potentials, which are ruled out, or severely constrained in the cold inflation scenario, but not necessarily in warm inflation, is provided. Next, the quantum field theory aspects in realizing explicit microscopic models for warm inflation are given. This includes the derivation of dissipation coefficients relevant in warm inflation for different particle field theory models. The dynamics of cosmological perturbations in warm inflation are then described. The general expression for the curvature scalar power spectrum is shown. We then discuss in detail the relevant regimes of warm inflation, the weak and strong dissipative regimes. We also discuss the results predicted in these regimes of warm inflation and how they are confronted with the observational data. We explain how the dissipative dynamics in warm inflation can address several long-standing issues related to (post-) inflationary cosmology. This includes recent discussions concerning the so-called swampland criteria and how warm inflation can belong to the landscape of string theory.

**Keywords:** inflationary cosmology; warm inflation; dissipative effects; cosmological perturbations





## 1. Introduction

Among the different proposals that attempted to implement consistent inflationary dynamics within an explicit quantum field theory realization, the warm inflation (WI) paradigm [1–3] is one of the most attractive. Warm inflation explores the fact that the inflationary dynamics is inherently a multifield problem, since the vacuum energy that drives inflation eventually must be converted to radiation, which generally comprises a variety of particle species. Thus, WI model realizations explore those associated dissipative processes to realize radiation production concurrently with the inflationary expansion[1]. This is the opposite of the more usual scenario of cold (supercooled) inflation (CI) [6–10], where a separated period of radiation production after the end of inflation (graceful exit) is required.

From a model building perspective, the recent developments have aimed at overcoming some of the important issues found in earlier particle physics realizations of warm inflation. In order to be able to sustain a nearly-thermal bath during WI, a sufficiently strong dissipation is typically required, such that some of the energy density in the inflaton can be converted to radiation. For this to happen, earlier particle physics realizations of WI required large field multiplicities [5,11]. These large field multiplicities can be difficult to be generated in simple models while keeping perturbativity and unitarity in these models [12] (see, however, for natural realizations in the context of brane models [13], or

in extra-dimensional models with a Kaluza–Klein tower [14]). One other difficulty in WI model building is to properly control both quantum and thermal corrections to the inflaton such as not to spoil the flatness of its potential, which could otherwise prevent inflation to happen. Earlier models for WI have made use, for instance, of both supersymmetry and heavy intermediate fields coupled to the inflaton for this purpose [4,15]. More recent implementations of WI have focused, instead, in using symmetry properties such as to be able to efficiently control the corrections to the inflaton potential [16–18]. Finally, from an effective field theory point of view, WI constructions that can be able to achieve strong dissipative regimes been shown to display quite appealing features. For example, already in some of the first studies in WI [19,20], it has been claimed that WI in the strong dissipative regime can also prevent super-Planckian field excursions for the inflaton, thus making WI potentially attractive in terms of an effective field theory consistent with an UV-complete realization in terms of quantum gravity [21–34]. Finally, the dissipative effects in WI can lower the energy scale of the inflaton and, as a result of this, the tensor-to-scalar ratio can be decreased with respect to what it would be in CI for the same type of primordial inflaton potential. This makes several primordial inflaton potential models that would otherwise be and had been discarded in CI, to be in line with the CMB observations in the context of WI [35]. The above are just a few examples of recent developments in WI and which have been attracting increasing interest in this intriguing alternative picture of inflation. In this paper, we review some of these major developments achieved in the area in the recent years.

This paper is divided as follows. In Section 2, we start by briefly reviewing the WI background dynamics and contrasting it to the CI picture. We discuss how a supplementary friction term in the Klein–Gordon equation is able to bring about a richer dynamics for WI. The smooth connection of the end of WI with the radiation-dominated regime is discussed. We show that there are several different possibilities for graceful exit depending on the form of the inflaton potential, the dissipation coefficient and whether being in the weak or strong dissipative regimes. In Section 3, we describe the necessary tools for calculating dissipation coefficients in the context of non-equilibrium quantum field theory and which are applied to WI. Some of the most recent microscopic realizations of WI are discussed and the respective derivation of the dissipation coefficients for these models is outlined. In Section 4, we discuss the cosmological perturbation theory for WI. Several important issues are discussed and the general derivation of the scalar of curvature power spectrum in WI is given. The bispectrum and non-Gaussianities in WI are also discussed. In Section 5, we discuss the observational constraints and other applications of the WI dissipative dynamics. It is shown how the dissipative particle production addresses/alleviates some of the long-outstanding problems in cosmology, e.g., related to the inflationary and post-inflationary phases and which CI cannot directly answer. In Section 5, we also discuss the connections which WI recently made with the so-called swampland criteria. We start by briefly reviewing the motivation behind the swampland conjectures [36–41]. We discuss why the dynamics of WI allows it to satisfy the swampland conjectures. Given the constraints imposed by the swampland conjectures, we find under which conditions WI is able to simultaneously satisfy the swampland conjectures and the implications of this for building inflationary models in string theory in the context of WI. An overview of different WI implementations and applications, including in the context of non-canonical models, is also given. Finally, in Section 6, we give our concluding remarks.

## 2. Background Dynamics of WI

A WI regime is typically realized when the inflaton field is able to dissipate its energy into other light degrees of freedom with a rate that is faster than the Hubble expansion. Thus, the produced particles have enough time to thermalize and become radiation. During this time, where the inflaton is decaying into the radiation particles and that can subsequently thermalize, one can then model their contributions as simply a radiation fluid with $\rho_r = \pi^2 g_* T^4 / 30$, with $\rho_r$, $T$, and $g_*$ being the radiation energy density, the temperature,

and the effective number of relativistic degrees of freedom of the produced particles. Hence, the total energy density of the universe in the WI scenario contains both the inflaton field and a primordial radiation energy density, i.e., $\rho = \rho_\phi + \rho_r$, where $\rho_\phi$ is the inflaton field energy density. Energy conservation then demands that the energy lost by the inflaton field must be gained by the radiation fluid. Therefore, the evolution equations can be obtained from the conservation of the energy-momentum tensor $T^{\mu\nu}$ [42],

$$\nabla_\mu T^{\mu\nu} = 0. \tag{1}$$

We work in the spatially flat Friedmann–Lemaître–Robertson–Walker (FLRW) metric, $ds^2 = -dt^2 + a(t)^2 \delta_{ij} dx^i dx^j$, where $a(t)$ is the scale factor. Hence, Equation (1) leads to a set of continuity equations for each component of the cosmological fluid,

$$\dot{\rho}_\alpha + 3H(\rho_\alpha + p_\alpha) = Q_\alpha, \tag{2}$$

where a dot here means a derivative with respect to the cosmic time $t$, with $\rho_\alpha$ and $p_\alpha$ being the energy density and pressure for each fluid component $\alpha$, respectively, and $H \equiv \dot{a}/a = \sqrt{\rho/3}/M_{\rm Pl}$ is the Hubble expansion rate. Here, $M_{\rm Pl} = (8\pi G)^{-\frac{1}{2}} \simeq 2.44 \times 10^{18}$ GeV is the reduced Planck mass and $G$ is Newton's gravitational constant. Moreover, $Q_\alpha$ in Equation (2) is a the source term, which describes the energy conversion between the species $\alpha$ accounted in the theory. The conservation of energy assures that $\sum_\alpha Q_\alpha = 0$. Therefore, the conversion of the inflaton energy density into radiation energy density in the WI scenario is, hence, described by the following set of equations [5]

$$\dot{\rho}_\phi + 3H(\rho_\phi + p_\phi) = -\Upsilon(\rho_\phi + p_\phi), \tag{3}$$
$$\dot{\rho}_r + 3H(\rho_r + p_r) = \Upsilon(\rho_\phi + p_\phi), \tag{4}$$

where $\Upsilon$ is the dissipation coefficient, which can generally be a function of the inflaton field $\phi$ and temperature $T$ and whose functional form depends on how WI is being described in terms of the microscopic physics [11,12,16–18]. Considering the energy density and pressure for a standard canonical inflaton field, i.e., $\rho_\phi = \dot{\phi}^2/2 + V(\phi)$ and $p_\phi = \dot{\phi}^2/2 - V(\phi)$, with $p_r = \rho_r/3$, Equations (3) and (4) reduce to

$$\ddot{\phi} + (3H + \Upsilon)\dot{\phi} + V_\phi = 0, \tag{5}$$
$$\dot{\rho}_r + 4H\rho_r = \Upsilon\dot{\phi}^2, \tag{6}$$

where $V_\phi$ is the derivative of the inflaton potential with respect to $\phi$. Although inflation happens when the energy density is dominated by the inflaton field potential $V$, i.e., $\rho_r$, $\dot{\phi}^2/2 \ll V$, such that the radiation energy density is sub-dominant, even so the produced radiation energy density can still satisfy $\rho_r^{1/4} > H$. Assuming thermalization, this condition then translates into $T > H$, which is usually considered as a condition for WI to happen. This condition is easy to understand. Since the typical mass for the inflaton field during inflation is $m_\phi \simeq H$, hence, when $T > H$, thermal fluctuations of the inflaton field will become important. Looking at Equation (5), one can immediately see that dissipative particle production effects manifest as an extra friction term in the equation of motion for the inflaton. Therefore, radiation will not be necessarily redshifted during inflation, because it can be continuously fed by the inflaton through dissipation. As a consequence, this can result in a sustainable quasi-stationary thermal bath during the inflationary dynamics. Such radiation production also results in entropy production. The entropy density $s$ is related to the radiation energy density by $Ts = 4\rho_r/3$, i.e., it is related to the temperature as $s = 2\pi^2 g_* T^3/45$, where we have considered a thermalized radiation bath as it is typically the case in the WI scenario. Then, Equation (6) can be rewritten in terms of the entropy density as follows [43]:

$$T(\dot{s} + 3Hs) = \Upsilon\dot{\phi}^2. \tag{7}$$

With the inflaton's potential dominating, an inflationary phase sets in. Thus, to solve Equations (5) and (6), one can use the so-called slow-roll approximation, which consists in dropping the leading derivative term in each equation, i.e., $\ddot{\phi} \ll 3H\dot{\phi}, V_\phi$ and $\dot{\rho}_r \ll 4H\rho_r, Y\dot{\phi}^2$. Hence, the slow-roll equations read as follows,

$$3H(1+Q)\dot{\phi} + V_\phi \simeq 0, \tag{8}$$

$$\rho_r \simeq \frac{3}{4}Q\dot{\phi}^2, \tag{9}$$

and $H \simeq \sqrt{V/3}/M_{\text{Pl}}$. In Equations (8) and (9), we have introduced $Q$,

$$Q = \frac{Y}{3H}, \tag{10}$$

which defines the dissipation ratio in WI and it measures the strength of the dissipative particle production effects in comparison to the spacetime expansion. There are two different regimes in WI that can be defined depending on the value of the dissipation ratio $Q$. The *weak dissipative regime* is when $Q < 1$. In this regime, dissipation is not expected to modify significantly the background dynamics. Hence, in this case, the background dynamics is similar as in CI. However, as we will see in the Section 4, thermal fluctuations of the radiation energy density can still strongly affect the field fluctuations, and also the primordial spectrum of perturbations as long as $T > H$. The other regime of WI is the *strong dissipative regime*, $Q > 1$. In this case, dissipation strongly modifies both the background dynamics and the primordial fluctuations. Note that $Q$ is not necessarily constant. In fact, it can increase or decrease depending on the form of the dissipation coefficient and inflaton potential. Therefore, there is also a possibility that a model can start in the weak dissipative regime, but later on to transit into the strong dissipative regime, or vice versa. Let us discuss in more details these different possibilities that can appear in WI and how dissipation ultimately affects the dynamics.

From the slow-roll equations, one can express the Hubble slow-roll parameter $\epsilon_H$ in terms of the so-called potential slow-roll parameter $\epsilon_V$ as follows,

$$\epsilon_H \equiv -\frac{\dot{H}}{H^2} \simeq \frac{1}{2}M_{\text{Pl}}^{-2}(1+Q)\frac{\dot{\phi}^2}{H^2} \simeq \frac{\epsilon_V}{(1+Q)}, \tag{11}$$

where $\epsilon_V = M_{\text{Pl}}^2(V_\phi/V)^2/2$. To reach the second equality in Equation (11), we have used the second Friedmann equation, i.e., $-2M_{\text{Pl}}^2\dot{H} = \rho + P$, together with Equation (9) and then used Equation (8) to reach the last equality. One can realize from Equation (11) that inflation will end when $\epsilon_H \simeq 1$ or, equivalently, when $\epsilon_V \simeq 1 + Q$. Moreover, looking at the last equality in Equation (11), one can immediately see that the equality between the Hubble slow-roll and the potential slow-roll parameters as observed in the CI, $\epsilon_H \simeq \epsilon_V$, does not hold in the WI scenario. In the CI scenario, the inflationary phase occurs when the Hubble slow-roll parameter is smaller than unity, i.e., $\epsilon_H \ll 1$, which means that the potential slow-roll parameter should also be smaller than unity. However, in the WI scenario, the inflationary phase occurs even when the potential slow-roll parameter is bigger than one (or even much bigger than one), provided that the dissipation ratio is large enough. This in particular alleviates the need for very flat potentials, as far as the background dynamics is concerned. Moreover, defining the number of e-folding as $N = \ln a$, i.e., $dN = Hdt = (H/\dot{\phi})d\phi$, which measures the expansion of the universe, one can see from the last two expressions in Equation (11) that $d\phi/dN = M_{\text{Pl}}\sqrt{2\epsilon_V}/(1+Q)$, which means that the inflaton field excursion can be much smaller in the WI scenario than in comparison to the CI scenario for the same variation of $\epsilon_H$. In fact, the inflaton field excursion in WI can be sub-Planckian even for very steep potentials. We will discuss later in Section 5 how such novel background dynamics will allow WI to reside in the landscape of the string theory.

When using the slow-roll approximation in Equations (3) and (4), one should carefully consider its consistency. This consistency check can be performed, for instance, using a linear stability analysis to determine under which conditions the system remains close to the slow-roll solution for many Hubble times [43]. Through this procedure, one finds that the sufficient conditions for the slow-roll approximation to hold are[2]

$$\epsilon_V, \eta_V, \beta_V \ll 1 + Q, \qquad 0 < b \ll \frac{Q}{1+Q}, \qquad |c| < 4, \tag{12}$$

where we have defined the additional quantities $\eta_V = M_{\mathrm{Pl}}^2 V_{\phi\phi}/V$, $\beta_V = M_{\mathrm{Pl}}^2 V_\phi \Upsilon_\phi/(V\Upsilon)$, $b = T V_{\phi T}/V_\phi$ and $c = T \Upsilon_T/\Upsilon$. The condition on the parameter $b$ states that the thermal correction to the inflaton potential should be small, while the condition on the parameter $c$ reflects the fact that radiation has to be produced at a rate larger than the red-shift due to the expansion of the universe. Therefore, slow-roll WI occurs when all conditions in Equation (12), together with $T > H$, are satisfied. However, one may wonder that the condition for WI to happen is in conflict with the conditions for the slow-roll approximation. Using Equation (9) and given that $T > H$ during inflation, one finds that

$$\frac{\dot{\phi}^2/2}{V(\phi)} > \frac{\pi^2 g_*}{135} Q^{-1} \frac{H^2}{M_{\mathrm{Pl}}^2}. \tag{13}$$

Since $H \ll M_{\mathrm{Pl}}$ in most inflationary models, one can find that the WI scenario is consistent with the slow-roll approximation even for the weak dissipation regime, $Q < 1$. Moreover, one shall further show that the radiation energy density will never exceed the potential energy in the slow-roll regime, thus guaranteeing a period of accelerated expansion. To this end, one can calculate the radiation energy density to potential energy density ratio as follows,

$$\frac{\rho_r}{V} \simeq \frac{1}{2} \frac{\epsilon_V}{1+Q} \frac{Q}{1+Q}. \tag{14}$$

During inflation $\epsilon_H \ll 1$, meaning that $\rho_r \ll V$, even for a large dissipation ratio, while at the end of inflation, $\epsilon_H \simeq 1$, i.e., $\epsilon_V \simeq 1 + Q$, implying that $\rho_r \simeq V$, if the strong dissipation regime can be achieved. Therefore, radiation will not be diluted and can even become dominant at the end of inflation. As a consequence, the universe can smoothly enter into the radiation-dominated epoch without the need of a separate reheating phase as required in the CI scenario. Therefore, there is a possibility that even potentials without a minimum can also be embedded into a WI scenario without any difficulty. In other words, those inflationary potentials without minimum, which have attracted considerable attention due to recently proposed swampland conjectures inspired from string theory, usually result in an ever-lasting inflationary phase in the CI scenario and they require another mechanism for termination of inflation. However, inflation will end due to dissipative particle production in the WI scenario even if the inflaton potential has no minimum. Hence, larger classes of inflationary potentials can be embedded into the WI scenario due to its richer dynamics in comparison to the CI case.

Having discussed under which conditions a slow-roll WI dynamics can be consistently achieved, one next consistency check is to investigate under which conditions the inflationary phase can end in this context. Looking at Equation (11), one can see that there are several possibilities for graceful exit in the WI scenario. The end of WI depends on the form of the potential, on the dissipation coefficient and whether the regime of weak or strong dissipation has been achieved. In other words, the inflationary phase can continue as long as $\epsilon_V < 1 + Q$ and it will end when $\epsilon_V \simeq 1 + Q$. Although in the CI scenario, inflation ends when $\epsilon_V$ increases during inflation, in the WI scenario, there is a possibility that even potentials with constant and decreasing $\epsilon_V$ have graceful exit, since $Q$ is also a dynamical parameter. In fact, depending on the evolution of $\epsilon_V$ and $Q$, there are generally three possibilities for graceful exit in the WI scenario. First, if $\epsilon_V$ increases, inflation ends

when $Q$ decreases, remains constant or not increases faster than $\epsilon_V$. In fact, potentials such as the monomial potentials, hilltop potentials, natural inflaton potential and the Starobinsky potential, which have a graceful exit in the CI scenario, also have graceful exit in WI depending on the form of the dissipation coefficient. Second, if $\epsilon_V$ is constant, in the exponential type of potentials, inflation ends only when $Q$ is a decreasing function. Third, if $\epsilon_V \gg 1$ and it is a decreasing function, inflation ends when $Q$ decreases much faster than $\epsilon_V$ and cross $\epsilon_V$ before it becomes less than unity. Although the last possibility exists in the WI scenario, it is very challenging from a model building point of view and these models usually do not end as in the case of the CI scenario (see [51] for more details). All of the aforementioned possibilities can be summarized in terms of the Hubble slow-roll parameter in such a way that inflation ends when $\epsilon_H$ increases with the number of e-folding. Therefore, taking the derivative with respect to the number of e-fold from Equation (11), one obtains the following inequality [51]

$$\frac{d \ln \epsilon_V}{d \ln N} > \frac{Q}{1+Q} \frac{d \ln Q}{dN}. \tag{15}$$

As long as the inequality (15) is satisfied, inflation will end. To understand under which conditions WI goes through graceful exit, one needs to find the evolution of $\epsilon_V$ and $Q$ during inflation. To this end, we need to fix both the dissipation coefficient and the potential function. Several different forms of dissipation coefficients were derived from first principles in quantum field theory during the development of the WI scenario. In the early development of WI, an inverse temperature-dependent dissipation coefficient, i.e., $\Upsilon \sim \phi^2/T$ was derived. However, soon after that, it was realized that such model suffers from large thermal corrections affecting the inflaton potential [52,53]. To overcome such difficulty, a two-stage mechanism was proposed (see, e.g., [4]) in which the thermal bath can be produced and sustained without introducing large thermal corrections to the inflaton potential. In this case, the dissipation coefficient has a cubic temperature dependence, i.e., $\Upsilon \sim T^3/\phi^2$. Later, in [16], a model with a dissipation coefficient with linear temperature dependence, i.e., $\Upsilon \sim T$ was also constructed and also a variant model [17] with an inverse temperature dependence in the high temperature regime to obtain $\Upsilon \sim T^{-1}$ was created. More recently, in [18], a dissipation coefficient with a simple cubic temperature dependence without field dependence, i.e., $\Upsilon \sim T^3$ was also realized. The models originating these forms of dissipation coefficients and others will be discussed in more details in Section 3. Almost all of the aforementioned dissipation coefficients can generally be parameterized as follows

$$\Upsilon(\phi, T) = C_\Upsilon T^c \phi^p M^{1-p-c}, \tag{16}$$

where $C_\Upsilon$ is a dimensionless constant that carries the details of the microscopic model used to derive the dissipation coefficient, such as the different coupling constants of the model (see, e.g., Section 3), $M$ is some mass scale in the model and depends on its construction, while $c$ and $p$ are numerical powers, which can be either positive or negative powers (note that the dimensionality of the dissipation coefficient in Equation (16) is $[\Upsilon] = [\text{energy}]$). Given this general form for the dissipation coefficient, one can find the dynamical evolution of the relevant parameters of the system, $\epsilon_V$, $Q$, $T/H$, and $T$ in terms of the potential slow-roll parameters as follows:

$$\frac{d\ln\epsilon_V}{dN} = \frac{4\epsilon_V - 2\eta_V}{1+Q}, \tag{17}$$

$$\frac{d\ln Q}{dN} = C_Q^{-1}[(2c+4)\epsilon_V - 2c\eta_V - 4p\kappa_V], \tag{18}$$

$$\frac{d\ln(T/H)}{dN} = C_Q^{-1}\left[\frac{7-c+(5+c)Q}{1+Q}\epsilon_V - 2\eta_V - \frac{1-Q}{1+Q}p\kappa_V\right], \tag{19}$$

$$\frac{d\ln T}{dN} = C_Q^{-1}\left(\frac{3+Q}{1+Q}\epsilon_V - 2\eta_V - \frac{1-Q}{1+Q}p\kappa_V\right), \tag{20}$$

where $\kappa_V = M_{Pl}^2 V_\phi/(\phi V)$ and $C_Q = 4 - c + (4+c)Q$ is a positive quantity, since Q is always positive and $-4 < c < 4$ from stability conditions [43–45].

For illustration purposes, in Figure 1, we show the space of parameters leading to different scenarios in WI when considering monomial potentials,

$$V(\phi) = \frac{V_0}{n}\left(\frac{\phi}{M_{Pl}}\right)^n, \tag{21}$$

and for different parametric forms for the dissipation coefficient in the weak dissipative regime. Using Equations (17) and (18) and the inequality (15), we find for which values in the space of parameters $(n, c, p)$, WI has a graceful exit. One should note that the conditions for WI to have a graceful exit are independent of being in the weak or in the strong dissipation regime as it is clear from Equation (18). Moreover, we also used Equations (19) and (20) to see how $T/H$ and $T$ evolve in the region for which WI has a graceful exit.

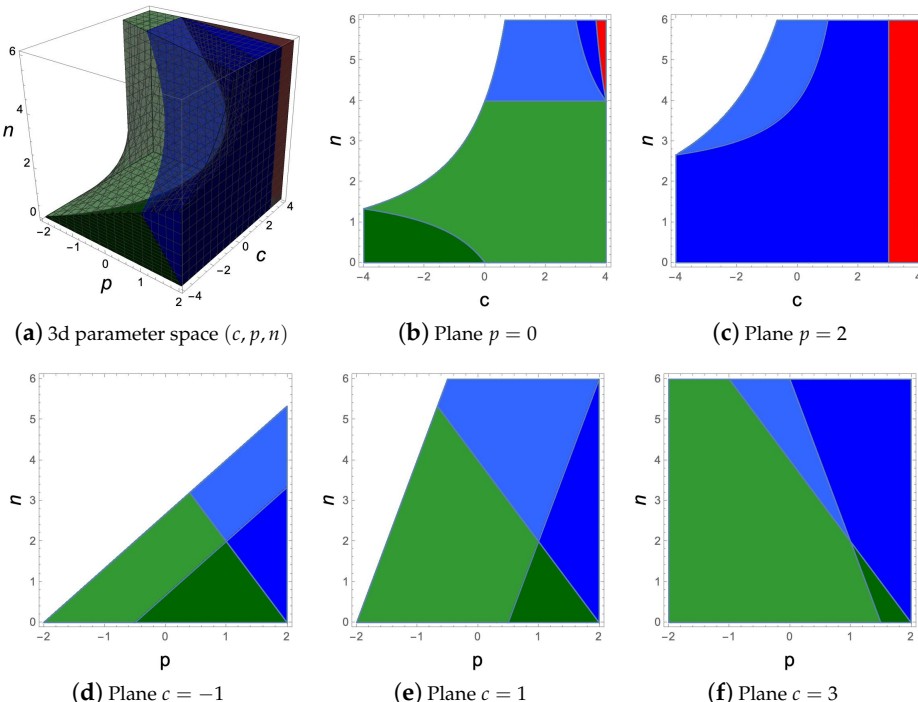

**(a)** 3d parameter space $(c, p, n)$     **(b)** Plane $p = 0$     **(c)** Plane $p = 2$

**(d)** Plane $c = -1$     **(e)** Plane $c = 1$     **(f)** Plane $c = 3$

**Figure 1.** The parameter space of WI in the weak dissipative regime ($Q < 1$) and for the monomial class of inflaton potentials. The shaded areas indicate WI has a graceful exit, while empty (white) space indicates where there is no graceful exit. The shaded areas are classified based on the behavior of $Q$, $T/H$, and $T$ during inflation when all three increase (light green), $Q$ decreases and both $T/H$ and $T$ increase (dark green), $T$ decreases and both $Q$ and $T/H$ increases (light blue), $T/H$ increases and both $Q$ and $T$ decreases (dark blue), and all three decrease (dark red). Figure based on [51].

In Figure 2, we also illustrate the space of parameters for WI in the strong dissipation regime to allow to see how the region for evolution for $T/H$ and $T$ changes. As we have discussed previously, $Q$ is a dynamical parameter, hence WI may start in the weak dissipative regime and be able to end in the strong dissipative regime, or vice versa. Therefore, $T/H$ and $T$ should not necessarily monotonically increase or decrease during inflation.

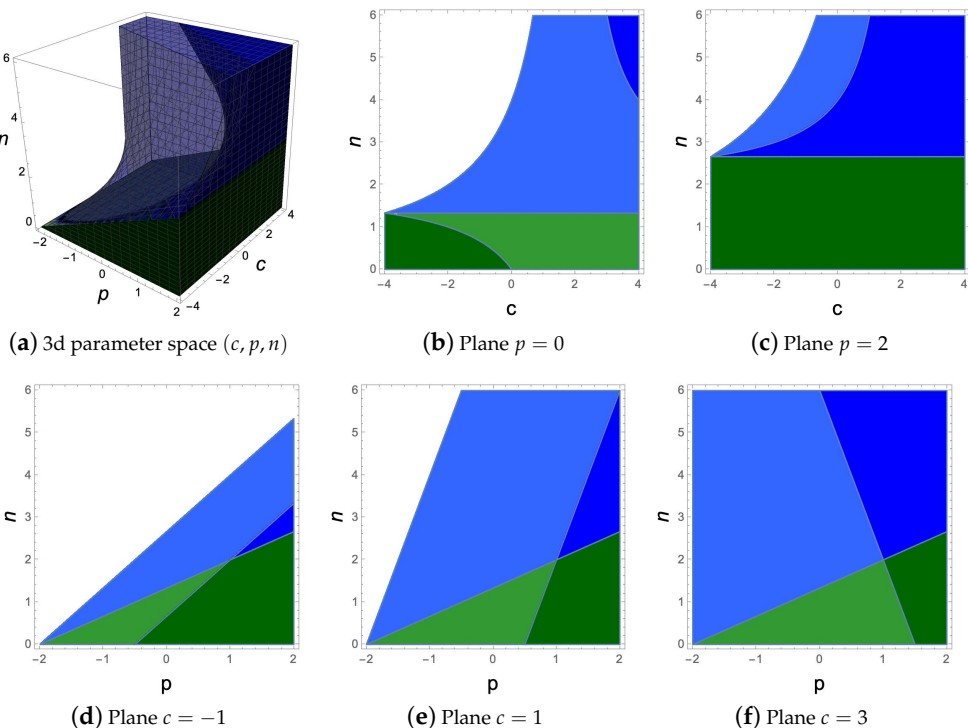

(**a**) 3d parameter space $(c, p, n)$     (**b**) Plane $p = 0$     (**c**) Plane $p = 2$

(**d**) Plane $c = -1$     (**e**) Plane $c = 1$     (**f**) Plane $c = 3$

**Figure 2.** The parameter space of WI in the strong dissipative regime ($Q > 1$) and for the monomial class of inflaton potentials. The shaded areas are classified based on the behavior of $Q$, $T/H$, and $T$ during inflation when all three increase (light green), $Q$ decreases and both $T/H$ and $T$ increase (dark green), $T$ decreases and both $Q$ and $T/H$ increase (light blue), and $T/H$ increases and both $Q$ and $T$ decrease (dark blue). Figure based on [51].

Comparing Figures 1 and 2, one may easily observe that there are possibilities for which $T/H$ and $T$ first increase and then decrease, or vice versa, depending on being in the weak or strong dissipation regime and $Q$ increases/decreases.

In Figure 3, for illustration purposes, the dynamical parameters of WI for a quadratic potential and linear temperature-dependent dissipation coefficient are plotted versus the number of e-folding. One can obviously see that the dissipation ratio $Q$, the temperature $T$, and the ratio $T/H$ all increase during the inflationary phase, as it was expected from Figure 1. Moreover, the radiation to potential energy density increases and becomes order unity shortly after the end of the inflation. That is because the condition $\epsilon_V \simeq 1 + Q$ is a weaker condition than $\epsilon_H = 1$ to specify the end of the inflationary phase in the WI scenario. In fact, the first condition points out that inflation ends when $\rho_r \simeq V/2$ while, in reality, one expects that inflation ends when the radiation energy density equals or suppresses the potential energy density as one can see in Figure 3. In this sense, $\epsilon_V \simeq 1 + Q$ predicts the end of inflation slightly earlier. In general, the weaker condition only underestimates the end of inflation by less than one e-folding. Thus, the condition $\epsilon_V = 1 + Q$ is still good enough as a way of estimating the instant where WI ends for all practical purposes. Furthermore, the inflaton field starts from super-Planckian values and ends sub-Planckian, with an overall super-Planckian field excursion, which is a typical feature of large field inflationary potentials and, in this case, for WI in the weak dissipative regime, as considered

in the example shown in Figure 3. Sub-Planckian field excursions in WI is possible in the strong dissipative regime of WI, as we will discuss later.

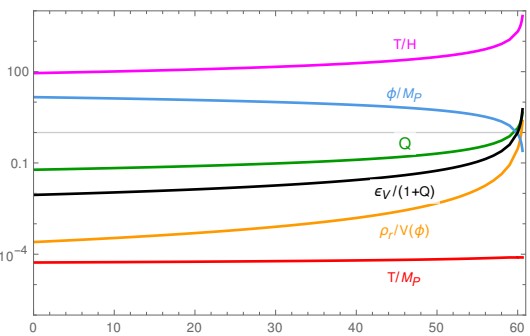

**Figure 3.** Evolution of the different quantities in the WI scenario for the case of $V(\phi) = \frac{1}{2}m^2\phi^2$, $Y = C_T T$, with $m = 10^{-7}M_{\text{Pl}}$ and $C_T = 0.002$.

As an additional example, in Figure 4, we plotted the space of parameters for another class of potential, hilltop-like potentials, given by

$$V(\phi) = V_0[1 - (\phi/\phi_0)^{2n}]^2, \tag{22}$$

with $n \geq 1$ and with the inflation taking place around the top (plateau) of the potential, $|\phi| \ll \phi_0$. We also consider that $\phi_0$ is sufficiently large such that inflation ends before the inflection point of the potential. Thus, we are considering that inflation takes place exactly in the concave part of the potential. Notice that here all the parameter space allows for a graceful exit, and $T$ and $T/H$ always increase for all space of parameters. Finally, it deserves to be noticed that exponential potentials such as

$$V(\phi) = V_0 \exp(-\alpha\phi/M_{\text{Pl}}), \tag{23}$$

can lead to a power law inflation only for $\alpha < \sqrt{2}$ in the CI scenario, while it does not have a graceful exit for $\alpha \geq \sqrt{2}$ [54]. However, looking at Equation (11), one can easily observe that the exponential potential not only can result in an accelerated expansion even for $\alpha > \sqrt{2}$, but also it has a graceful exit as long as the dissipation ratio is decreasing during inflation, i.e., when $c > 2$, regardless of the value of $p$ for $\alpha\phi/M_{\text{Pl}} > 1$. This case and other forms of primordial inflaton potentials have been studied in details in [51].

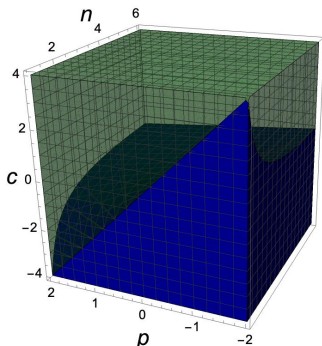
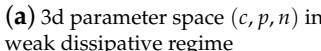
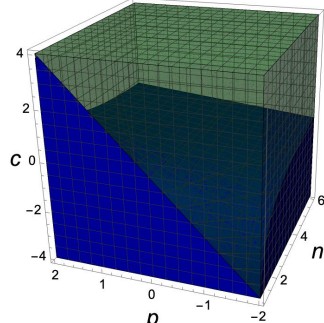

**(a)** 3d parameter space $(c, p, n)$ in weak dissipative regime

**(b)** 3d parameter space $(c, p, n)$ in strong dissipative regime

**Figure 4.** The parameter space in WI for the hilltop-like class of inflaton potentials. The shaded areas indicate that WI has a graceful exit, while empty (white) space indicates where there is no graceful exit. The shaded areas are classified based on the behavior of $Q$, $T/H$, and $T$ during inflation when all three increase (dark green), and $Q$ decreases and both $T/H$ and $T$ increase (dark blue). Figure based on [51].

### 3. Deriving Dissipation Coefficients in WI

The dynamics in WI is intrinsically a result of particle production processes able to happen during inflation. The generic idea is that as the inflaton evolves in time, moving around its potential, it might excite any other fields that are coupled to it. This in turn can produce relativistic particles and maintain a quasi-stationary radiation bath throughout the inflationary regime. At the end of the accelerated inflationary regime, the universe can then smoothly transit to the radiation-dominated regime.

Crucial to the idea of WI is then the role played by the dissipation coefficient Y, e.g., in the inflaton effective evolution equation, Equation (5). As in any inflaton model, we expect the inflaton to be necessarily coupled to other fields, which similar to the case of CI are required for reheating, the energy density in the inflaton will eventually be transferred to radiation. The emergence of dissipative processes in this case is reminiscent of the so-called Caldeira and Leggett-type of models [55]. There is a relevant degree of freedom, in which we are interested in the dynamics and which here is the background inflaton field. The system (the inflaton background) is in turn coupled to other degrees of freedom, i.e., any other fields coupled to the inflaton, which are regarded as environmental degrees of freedom. Below, we sketch the general idea for completeness, but which can also be found in details in many other previous papers, in particular, in [4,52,53,56,57]. We can describe the inflaton field and environment fields through a generic Lagrangian density of the form:

$$\mathcal{L}[\Phi, X, Y] = \mathcal{L}[\Phi] + \mathcal{L}[X] + \mathcal{L}[Y] + \mathcal{L}_{\text{int}}[\Phi, X] + \mathcal{L}_{\text{int}}[X, Y], \tag{24}$$

where here $\Phi$ is the inflaton field, or more specifically, its zero mode, represented by the homogeneous background value, while $X$ is describing any field or degrees of freedom coupled directly to the inflaton field (such as fermion fields or other bosons that can be scalars or vector fields), while $Y$ can be any other fields not necessarily coupled to the inflaton, such as additional bosons and/or fermions, but that can be coupled to $X$. The different interactions are described by the terms in Equation (24), $\mathcal{L}_{\text{int}}[\Phi, X]$, for the interactions of the inflaton with the $X$ fields, and $\mathcal{L}_{\text{int}}[X, Y]$, for the interactions between $X$ and $Y$, but not directly with the inflaton field.

The evolution of the inflaton field is then determined from Equation (24) by integrating over the environment fields $X$ and $Y$. This can be done, for example, in the context of the in-in Schwinger closed-time path functional formalism (for a textbook account, see, for example [58]) or, equivalently, through the influence functional formalism [59]. The formal expression for the evolution for a background field value $\Phi(x)$ turns out to be in the form of a stochastic equation of motion of the form [4,52,53]

$$\partial^2 \Phi + V'_{\text{eff}}(\Phi) + \int d^4x' \mathcal{D}^{(X)}(x, x')\dot{\Phi}_c(x') = \xi(x), \tag{25}$$

where $V'_{\text{eff}}(\Phi) = dV_{\text{eff}}(\Phi)/d\Phi$, with $V_{\text{eff}}(\Phi)$ being the effective potential for $\Phi$. The term with $\mathcal{D}^{(X)}(x, x')$, defined as

$$\Sigma_\rho^{(X)}(x, x')\text{sgn}(t - t') = -\frac{\partial}{\partial t'}\mathcal{D}^{(X)}(x, x'), \tag{26}$$

describes the dissipative effects due to the interaction of the inflaton with the environment fields, while $\xi(x)$ represents stochastic noise fluctuations, which affect the dynamics of $\Phi_c$ and describe Gaussian processes with the general properties of having zero mean, $\langle \xi(x) \rangle = 0$, and two-point correlation function given by

$$\langle \xi(x)\xi(x') \rangle = \Sigma_F^{(X)}[\Phi_c](x, x'). \tag{27}$$

In Equations (26) and (27), $\Sigma_\rho^{(X)}$ and $\Sigma_F^{(X)}$ are the self-energy terms in the Schwinger–Keldysh real time formalism of quantum field theory [59]. Furthermore, both noise and dissipation terms are related to each other through a generalized fluctuation dissipation

relation [4], as expected for a Langevin-like evolution describing a stochastic process under both dissipation and noise terms.

If the field $\Phi$ is slowly varying on the response timescale $1/\Gamma$, with the $\Gamma$ derived from the self-energies terms coming from the $X$ and $Y$, then $\dot{\Phi}/\Phi \ll \Gamma$. Under these conditions, which are typically referred to as the adiabatic approximation, a simple Taylor expansion of the non-local terms in Equation (25) can be performed. Furthermore, if $\Gamma > H$, the produced radiation through the dissipative processes can be thermalized sufficiently fast. Under the further adiabatic condition that $\Gamma \gg \dot{T}/T$, it is expected that the produced radiation can be maintained in a close to thermal equilibrium state at a temperature $T$. These conditions provide a clear separation of timescales in the system, which allows to approximate Equation (25) in the form of a local, Markovian equation of motion [60], with a local dissipation coefficient $\Upsilon$ defined by

$$\Upsilon = \int d^4x' \, \Sigma_R[\Phi](x, x') \, (t' - t), \tag{28}$$

where $\Sigma_R^{(X)}(x, x') = \Sigma_\rho^{(X)}(x, x')\theta(t - t')$ is a retarded self-energy term. Overall, Equation (25), when working in a Friedmann–Lemaître–Robertson–Walker (FLRW) background metric, then becomes of the form

$$\left[\frac{\partial^2}{\partial t^2} + (3H + \Upsilon)\frac{\partial}{\partial t} - \frac{1}{a^2}\nabla^2\right]\Phi + \frac{\partial V_{\text{eff}}(\Phi)}{\partial \Phi} = \xi_T, \tag{29}$$

where $\xi_T$ describes thermal (Gaussian and white) noise fluctuations in the local approximation as defined above and it is connected to the dissipation coefficient through a Markovian fluctuation–dissipation relation,

$$\langle \xi_T(\mathbf{x}, t)\xi_T(\mathbf{x}', t')\rangle = 2\Upsilon T a^{-3}\delta(\mathbf{x} - \mathbf{x}')\delta(t - t'), \tag{30}$$

where the average here is to be interpreted as been taken over the statistical ensemble. Stochastic terms can also be ascribed for quantum contributions and will be important when deriving the perturbations equations. This will be discussed in the next section, Section 4.

The effective evolution equation for the inflaton, Equation (5), then follows from the localized form for the effective equation of motion, Equation (29) by separating the background field into its homogeneous term $\phi(t)$. On the other hand, the non-homogeneous fluctuations over $\phi(t)$, denoted as $\delta\phi(\mathbf{x}, t)$, will describe the fluctuations of the inflaton and to be considered in the density perturbation equations (see next section). The latter are explicitly dependent on the stochastic noise term. From the above definitions, once a particular interaction Lagrangian density is given, the corresponding dissipation term $\Upsilon$ can be computed explicitly. Various examples of interactions and the resulting dissipation terms have been derived and given, e.g., in [61].

Below, we summarize some of the most important microscopic particle physics constructions considered for WI. As already explained before, the particle physics implementation for WI seeks models for which a significant dissipation can be generated during inflation and, at the same time, the radiative and thermal corrections for the inflaton potential are kept under control, such that the flatness of the inflaton potential is not spoiled. The first particle physics model constructions in WI achieved this using quantum field theory models in supersymmetry (SUSY). This is the case of the first two models described below.

### 3.1. The Distributed Mass Model

In the distributed mass model (DMM) [3], there are a set of scalar $\chi_j$ and fermionic $\psi_j$ fields coupled to the inflaton through a series of couplings of the form $g^2(\phi - M_j)^2\chi_j^2$ and $h(\phi - M_j)\bar{\psi}_j\psi_j$, respectively. The idea is that as the inflaton is evolving, eventually the inflaton satisfies $\phi \sim M_j$. At this point, the masses of the fields coupled to it can become very

light. In particular, as these masses become smaller than the temperature, those fields will get thermally excited and the inflaton will be able to decay into those fields. A sequence of mass distributions $M_j$ can then be constructed in such a way that as the background inflaton field evolves, it is able to dissipate energy into these fields throughout the inflationary regime. This process can then be described by a dissipative term in the inflaton evolution equation [53]. The idea is reminiscent of string theory model constructions [62]. In this case, the inflaton is interpreted as an excited string zero mode. This string mode can then be interacting with massive string levels. As the string levels can be highly degenerate, a distribution of mass states can emerge as a type of fine structure splitting of those levels. The DMM realization for WI was also revised more recently in [63]. Both radiative and thermal corrections coming from the large set of bosonic and fermionic fields coupled to the inflaton can be well controlled by constructing the DMM in the context of SUSY. Despite the inflaton background field and thermal effects both break supersymmetry, there are still large cancellations between fermion and boson contributions [64]. The DMM can be implemented by a superpotential in the context of SUSY, given by

$$W = \sum_j \left[ \frac{g}{2}(\Phi - M_j)X_j^2 + \frac{h}{2}X_jY_j^2 \right], \tag{31}$$

where $\Phi$, $X_j$, and $Y_j$ chiral superfields, with (complex) scalar and fermion components $\varphi$ and $\psi_\varphi$ for $\Phi$, $\chi_j$ and $\psi_{\chi_j}$ for $X_j$, and $\sigma_j$ and $\psi_{\sigma_j}$ for $Y_j$. The inflaton is associated with the real component of $\varphi$. The $g$ and $h$ are coupling constants and the sum is taken over an arbitrary distribution of supermultiplets $X_j$ and $Y_j$. From Equation (31), we can derive the scalar $\mathcal{L}_S$ and fermionic $\mathcal{L}_F$ Lagrangian density interaction terms, which are defined, respectively, as [64]

$$-\mathcal{L}_S = |\partial_\Phi W|^2 + \sum_j |\partial_{X_j} W|^2 + \sum_j |\partial_{Y_j} W|^2, \tag{32}$$

and

$$-\mathcal{L}_F = \frac{1}{2}\sum_{n,m} \frac{\partial^2 W}{\partial \xi_n \partial \xi_m} \bar{\psi}_n P_L \psi_m + \frac{1}{2}\sum_{n,m} \frac{\partial^2 W^\dagger}{\partial \xi_n^\dagger \partial \xi_m^\dagger} \bar{\psi}_n P_R \psi_m, \tag{33}$$

where $\xi_n$ is a superfield: $\Phi$, $X_j$, $Y_j$ and $P_L = 1 - P_R = (1 + \gamma_5)/2$ are the chiral projection operators acting on Majorana 4-spinors. Equations (32) and (33) then lead to the explicit Lagrangian density relevant contributions that are given by [63]

$$
\begin{aligned}
-\mathcal{L}_S \;=\; & g^2 \sum_j (\phi - M_j)^2 |\chi_j|^2 + \frac{gh}{2}\sum_j (\phi - M_j)\left[ \chi_j(\sigma_j^\dagger)^2 + \chi_j^\dagger \sigma_j^2 \right] \\
& + h^2 \sum_j |\chi_j|^2 |\sigma_j|^2 + \frac{g^2}{4}\sum_j |\chi_j|^4 + \frac{h^2}{4}\sum_j |\sigma_j|^4,
\end{aligned} \tag{34}
$$

and

$$
\begin{aligned}
-\mathcal{L}_F \;=\; & \frac{g}{2}\sum_j (\phi - M_j)\bar{\psi}_{\chi_j} P_L \psi_{\chi_j} + \frac{g}{2}\sum_j (\phi - M_j)\bar{\psi}_{\chi_j} P_R \psi_{\chi_j} \\
& + \frac{h}{2}\sum_j \chi_j \bar{\psi}_{\sigma_j} P_L \psi_{\sigma_j} + \frac{h}{2}\sum_j \chi_j^\dagger \bar{\psi}_{\sigma_j} P_R \psi_{\sigma_j} + h\sum_j \sigma_j \bar{\psi}_{\sigma_j} P_L \psi_{\chi_j} + h\sum_j \sigma_j^\dagger \bar{\psi}_{\sigma_j} P_R \psi_{\chi_j}.
\end{aligned} \tag{35}
$$

From the interactions terms in Equations (34) and (35), we can determine the dissipation coefficient $\Upsilon$, which will receive contributions both from the scalar bosons [3,53] (see also [63] for details)

$$\Upsilon^S(\phi, T) = \sum_{j=1}^{N_{th}} \frac{32g^4}{\pi h^2 \left(16\frac{m_{\chi_j}^2}{\tilde{m}_{\chi_j}^2} + \frac{\tilde{m}_{\chi_j}^2}{T^2}\right)} \ln\left(\frac{2T}{\tilde{m}_{\chi_j}}\right) \frac{(\phi - M_j)^2}{\tilde{m}_{\chi_j}},\tag{36}$$

and from the fermions [16],

$$\Upsilon^F(T) = \sum_{j=1}^{N_{th}} C_T^F T, \quad C_T^F \simeq \frac{3g^2}{h^2[1 - 0.34\ln(h)]},\tag{37}$$

where in the above equations, the sum runs over the number $N_{th}$ of thermally excited field modes and the masses $m_{\chi_j}$ and $\tilde{m}_{\chi_j}$ are defined, respectively, as $m_{\chi_j} = g(\phi - M_j)$ and $\tilde{m}_{\chi_j}^2 = m_{\chi_j}^2 + g^2 T^2/12 + h^2 T^2/8$. One notes that the dissipation coefficient in the DMM depends on the mass distribution $M_j$. Assuming [63] $M_j = \phi + j\Delta M(\phi, T, m, g)$, where $\Delta M$ denotes the mass gap in the tower of states, different functional forms for the dissipation coefficient in the DMM can be generated through different choices of $\Delta M(\phi, T, m, g)$. This motivates the parametric choice Equation (16) adopted in many works in WI.

### 3.2. The Two-Stage Mechanism Model

Without a way of controlling the thermal corrections of the radiation fields that are directly coupled to the inflaton, it is expected that the finite temperature of the radiation bath will induce large thermal corrections to the inflaton mass, leading to $m_\phi \sim T > H$. If this occurs, successful realizations of WI in the simplest models are jeopardized. In this case, the additional friction caused by the dissipation effects through $\Upsilon$ cannot overcome the increase in the inflaton's mass. In the DMM discussed above, even though there are couplings of the inflaton directly to the radiation fields, it is able to evade this problem by a judicious choice of the mass distribution function. However, in other model realizations, this is not a simple task to achieve, as discussed originally in [53,65]. However, we also recall that fields that are directly coupled to the inflaton will tend to acquire large masses during inflation due to the large background field value $\phi$. This then suggests that WI can be better implemented in scenarios where the inflaton does not couple directly to the radiation fields, but instead to heavy intermediate fields, which can be either bosons or fermions, with masses such that $m_\chi, m_\psi > T$. This naturally leads to thermal corrections to the inflaton potential that are Boltzmann suppressed. Furthermore, once again, we can use SUSY to control potentially large radiative corrections to the inflaton. In turn, the heavy fields can be made coupled to the radiation fields, which remain decoupled from the inflaton sector. Once again, as the inflaton dynamics change the masses of the heavy fields, these can decay into the light radiation fields. These processes provide a way through which the inflaton can dissipate its energy into radiation. A significant dissipation can be generated depending on the field multiplicities. This is the two-stage decay mechanism model for WI.

The two-stage mechanism for WI can be implemented through a supersymmetric model with chiral superfields $\Phi$, $X$ and $Y_i$, $i = 1, \ldots, N_Y$, and described by the superpotential

$$W = \frac{g}{2}\Phi X^2 + \frac{h_i}{2}XY_i^2 + f(\Phi),\tag{38}$$

where a sum over the index $i$ is implicit. The scalar component of the superfield $\Phi$ describes the inflaton field, with an expectation value $\phi/\sqrt{2}$, which we assume to be real, and the generic holomorphic function $f(\Phi)$ describes the self-interactions in the inflaton sector. The superpotential Equation (38) leads to the Lagrangian density describing the interactions between the inflaton field $\phi$ with the other scalars and fermions as given by [12]:

$$\mathcal{L}_{scalar} = V(\phi) + \frac{1}{2}g^2\phi^2|\chi|^2 + \frac{g}{2}\sqrt{V(\phi)}\left(\chi^2 + \chi^{\dagger 2}\right) + \frac{g^2}{4}|\chi|^4 +$$
$$+ \frac{h_i}{2}\frac{g\phi}{\sqrt{2}}\left(\chi\sigma_i^{\dagger 2} + \chi^\dagger\sigma_i^2\right) + \frac{h_i h_j}{4}\sigma_i^2\sigma_j^{\dagger 2} + h_i^2|\chi|^2|\sigma_i|^2 , \tag{39}$$

and

$$\mathcal{L}_{fermion} = \frac{g\phi}{\sqrt{2}}\bar{\psi}_\chi P_L\psi_\chi + h_i\chi\bar{\psi}_{\sigma_i}P_L\psi_{\sigma_i} + \frac{h_i}{2}\sigma_i\bar{\psi}_{\sigma_i}P_L\psi_\chi + \text{h.c.} , \tag{40}$$

where the scalar components of the superfields $X$ and $Y_i$ were denoted by $\chi$ and $\sigma_i$, respectively, the fermionic components by $\psi_\chi$ and $\psi_{\sigma_i}$, respectively, $V(\phi) = |f'(\phi)|^2$ is the potential driving the inflaton and $P_L = (1 - \gamma_5)/2$ is the left-handed chiral projector.

The leading dissipation coefficient obtained for this model and using the quantum field theory expression Equation (28), can be explicitly derived and it is given by [61]:

$$\Upsilon = \frac{4}{T}\left(\frac{g^2}{2}\right)^2\varphi^2\int\frac{d^4p}{(2\pi)^4}\rho_\chi^2 n_B(1 + n_B) , \tag{41}$$

where $n_B(p_0) = [e^{p_0/T} - 1]^{-1}$ is the Bose–Einstein distribution and $\rho_\chi$ is the spectral function for the $\chi$ field,

$$\rho_\chi(p_0, p) = \frac{4\omega_p\Gamma_\chi}{(p_0^2 - \omega_p^2)^2 + 4\omega_p^2\Gamma_\chi^2} , \tag{42}$$

with $\Gamma_\chi$ denoting the decay width for the heavy fields $\chi$, which includes contributions from both the bosonic and fermionic final states in the $Y_i$ multiplets, $\omega_p = \sqrt{\tilde{m}_\chi^2 + p^2}$ for modes of 3-momentum $|\mathbf{p}| = p$ and energy $p_0$, while the thermal mass for the $\chi$ fields is $\tilde{m}_\chi^2 = m_\chi^2 + h^2 N_Y T^2/8$ (where here we are assuming all couplings $h_i = h$) and $m_\chi = g\phi/\sqrt{2}$. For larger values of the mass and effective coupling, the main contribution to the dissipation coefficient comes from *virtual* $\chi$ modes with low momentum and energy, $p, p_0 \ll m_\chi$, such that one can use the approximation $(p_0^2 - \omega_p^2)^2 \simeq \tilde{m}_\chi^4$. In the case where these modes also have a narrow width and thermal mass corrections can be neglected, $\Gamma_\chi \ll \tilde{m}_\chi \sim m_\chi$, then the spectral function Equation (42) can be simply expressed as [4,61] $\rho_\chi \simeq 4\Gamma_\chi/m_\chi^3$. Under these circumstances, the dissipation coefficient $\Upsilon$ describing the dissipation mediated by the decay of virtual scalar modes $\chi \to$ light radiation fields, is given by [12]:

$$\Upsilon = C_\phi\frac{T^3}{\phi^2} , \qquad C_\phi \simeq \frac{1}{4}\alpha_h N_X , \tag{43}$$

for $\alpha_h = h^2 N_Y/4\pi \lesssim 1$ and $N_{X,Y}$ are the chiral multiplets.

To obtain sufficient dissipation in these models, it is usually required large values for the numbers of multiplets $N_{X,Y}$. While these can be seen as a possible drawback for this model, there are well-motivated scenarios where this can be naturally achieved, such as in [13] using brane constructions, or in [14], where large field multiplicities can be allowed due to a Kaluza–Klein tower in extra-dimensional scenarios.

Instead of relying on SUSY as a way to suppress radiative corrections, we can try to arrange for special interactions of the inflaton with the environment fields such as to be able to control the thermal corrections to the inflaton potential. This can also be used to avoid the large multiplicities that would otherwise be required to produce sufficiently large dissipation, such that WI can be realized. To avoid these complications, we can make use of other simpler models exploring well-motivated symmetry properties for the interactions. This is the case for the next three particle physics model constructions for WI to be discussed below.

### 3.3. The Warm Little Inflation Model

In the warm little inflation (WLI) model, first introduced in [16], the inflaton is assumed to be a pseudo-Nambu–Goldstone boson (PNGB) of a gauge symmetry that is collectively broken. The idea is reminiscent of the "Little Higgs" models of electroweak symmetry breaking [66–68], where the Higgs boson is also considered to be a PNGB coming from a collectively broken global symmetry. A feature displayed by the type of construction in these type of models is that the PNGB has a mass that is naturally protected against large radiative corrections (for a review, see, e.g., [69]).

The construction of the WLI model uses a very minimum set of ingredients. There are two complex scalar fields, $\phi_1$ and $\phi_2$, which share the same $U(1)$ symmetry. The scalar potential for them allow both fields to have a nonzero vacuum expectation value, $\langle \phi_1 \rangle \equiv M_1/\sqrt{2}$ and $\langle \phi_2 \rangle \equiv M_2/\sqrt{2}$, where $M_1$ can be taken to be equal to $M_2$ without loss of generality, $M_1 = M_2 \equiv M$. With both fields sharing the same Abelian charge, after the Higgs mechanism, only one the phases of the complex scalar fields (the Nambu–Goldstone boson), or a linear combination of them, is absorbed as the longitudinal component of the $U(1)$ gauge boson, providing mass to it. The other phase (or linear independent combination of the phases) remains, however, as a physical degree of freedom, becoming, e.g., a singlet. The two complex scalar fields, in the broken phase, can then be expressed in the form

$$\phi_1 = \frac{M}{\sqrt{2}}e^{i\phi/M}, \qquad \phi_2 = \frac{M}{\sqrt{2}}e^{-i\phi/M}. \tag{44}$$

The radial modes in Equation (44) decouple when $M \gtrsim T$ and the singlet $\phi$, the PNBG, is taken to be the inflaton. One notes that as such, we can assume for $\phi$ an arbitrary scalar potential that can be sufficiently flat to sustain the inflaton. Dissipation in the model comes from the coupling of $\phi_1$ and $\phi_2$ to other fields. For example, they can be coupled to left-handed fermions $\psi_1$ and $\psi_2$ with the same $U(1)$ charge of the scalar fields. The right-handed components of these fermion fields, $\psi_{1R}$ and $\psi_{2R}$, can be taken to be gauge singlets, in a construction similar as in the Glashow–Weinberg–Salam of the standard model of particle physics. A decay width for these fermions, which will contribute to the dissipation coefficient, can be generated by coupling them to additional Yukawa interactions, involving a scalar singlet $\sigma$ and chiral fermions $\psi_{\sigma R}$, carrying the same charge of $\psi_1$ and $\psi_2$, and $\psi_{\sigma L}$, with zero charge. For all these fields, we impose an interchange symmetry for the complex scalars $\phi_1$ and $\phi_2$, $\phi_1 \leftrightarrow i\phi_2$, and also for the fermions $\psi_1$ and $\psi_2$, $\psi_{1L,R} \leftrightarrow \psi_{2L,R}$. The overall interaction Lagrangian density is [16]

$$\mathcal{L}_{\text{int}} = \mathcal{L}_{\phi\psi} + \mathcal{L}_{\psi\sigma}, \tag{45}$$

with

$$\mathcal{L}_{\phi\psi} = -\frac{g}{\sqrt{2}}(\phi_1 + \phi_2)\bar{\psi}_{1L}\psi_{1R} + i\frac{g}{\sqrt{2}}(\phi_1 - \phi_2)\bar{\psi}_{2L}\psi_{2R}, \tag{46}$$

and

$$\mathcal{L}_{\psi\sigma} = -h\sigma \sum_{i=1,2} (\bar{\psi}_{iL}\psi_{\sigma R} + \bar{\psi}_{\sigma L}\psi_{iR}), \tag{47}$$

which respects the symmetries imposed on the model. Due to the symmetries, and using Equation (44), the masses for the fermions $\psi_1$ and $\psi_2$, $m_1$ and $m_2$, respectively, are then given by

$$m_1 = gM\cos(\phi/M), \quad m_2 = gM\sin(\phi/M), \tag{48}$$

and, hence, they remain always bounded, $m_{1,2} \leq gM$, and they can be arranged to remain light during inflation if $gM \lesssim T \lesssim M$. The advantage of the interactions in Equation (46) is that they lead to no quadratic corrections for $\phi$ in the inflaton potential, thus, the inflaton

mass does not receive neither radiative nor thermal corrections. The inflaton potential is only affected by subleading Coleman–Weinberg corrections, as it was demonstrated in [16].

With the above interactions, the dissipation coefficient for the WLI model becomes [16]

$$\Upsilon = C_T T, \qquad C_T \simeq \alpha(h) g^2/h^2, \tag{49}$$

where $\alpha(h) \simeq 3/[1 - 0.34 \ln(h)]$.

The model has also been shown to produce a well-motivated dark matter candidate [70]. This can be possible because soon after inflation, the fermions coupled to the inflaton field are no longer light. The dissipation coefficient Equation (49) becomes Boltzmann-suppressed. For an inflaton potential augmented by a quadratic term with a mass term $m_\phi$, then, when $H < m_\phi$, the inflaton field is underdamped and it will oscillate around the minimum of its potential. The equation of state oscillates around $w_\phi = 0$, and the energy density scales as $\rho_\phi \propto 1/a^3$. This is the same behavior as ordinary matter. Due to the symmetries of the model, the inflaton rarely decays and an abundance of inflaton energy density can remain in the coherent oscillating state. This makes the inflaton in the WLI model to be a possible valid dark matter candidate.

### 3.4. The Warm Little Inflation Model—Scalar Version

A variant [17] of the WLI model discussed above is when the complex scalar fields $\phi_1$ and $\phi_2$ are now coupled directly to two other complex scalar fields $\chi_1$ and $\chi_2$, instead of fermions. As in the previous model, it is also imposed the discrete interchange symmetry $\phi_1 \leftrightarrow i\phi_2$, $\chi_1 \leftrightarrow \chi_2$. The complex scalar fields $\chi_{1,2}$ can also have a Yukawa interaction to fermions, self-interact, and also interact between each other. Then, the interacting Lagrangian density can now be expressed as [17]

$$\mathcal{L}_{\text{int}} = \mathcal{L}_{\phi\chi} + \mathcal{L}_{\chi\psi}, \tag{50}$$

with

$$\mathcal{L}_{\phi\chi} = -\frac{1}{2}g^2|\phi_1 + \phi_2|^2|\chi_1|^2 - \frac{1}{2}g^2|\phi_1 - \phi_2|^2|\chi_2|^2, \tag{51}$$

and

$$\mathcal{L}_{\chi\psi} = \sum_{i \neq j = 1,2} \left( h\chi_i \bar{\psi}_L \psi_R + \text{h.c.} - \frac{\lambda}{2}|\chi_i|^4 - \lambda'|\chi_i|^2|\chi_j|^2 \right). \tag{52}$$

One notes that under the parameterization given by Equation (44), the masses for the scalar fields $\chi_1$ and $\chi_2$ are still of the same form as in Equation (48). Hence, they are still bounded and can be light with respect to the temperature just the same way as in the previous model. In the high temperature regime, $m_{1,2} \ll T$, the inflaton potential receives leading order thermal corrections of the form $m_1^2(\phi)T^2/12 + m_1^2(\phi)T^2/12 = g^2M^2T^2/12$ and, thus, the inflaton mass here also does not receive any thermal corrections, while also not receiving any important radiative contributions.

In this scalar field variant of the WLI model, the dissipation coefficient now becomes [17]

$$\Upsilon \simeq \frac{4g^4}{h^2} \frac{M^2 T^2}{\tilde{m}_\chi^3} \left[ 1 + \frac{1}{\sqrt{2\pi}} \left( \frac{\tilde{m}_\chi}{T} \right)^{3/2} \right] e^{-\tilde{m}_\chi/T}, \tag{53}$$

where we have taken the average of the oscillatory terms for field excursions $\Delta\phi \gg M$ and $\tilde{m}_\chi$ is the thermal mass for the $\chi_{1,2}$ fields, which, under the same average over the oscillatory terms, is the same for both scalars and given by $\tilde{m}_\chi^2 \simeq g^2M^2/2 + \alpha^2T^2$, where $\alpha^2 \simeq (h^2 + 2\lambda + \lambda')/12$ when taking the interactions in Equation (52). It is noticed that when considering the leading thermal contribution in $\tilde{m}_\chi$, i.e., $\tilde{m}_\chi \sim \alpha T$, the dissipation coefficient (53) varies with the temperature as $\Upsilon(T) \propto T^{-1}$, realizing the case with $c = -1$, $p = 0$ in Equation (16). On the other hand, as the temperature of the thermal bath drops and the vacuum term $gM/\sqrt{2}$ in $\tilde{m}_\chi$ starts no longer to be negligible, it effectively would correspond to values of $c > -1$, with a limiting case of $c = 2$ when $gM/\sqrt{2} \gg \alpha T$ and with

an exponentially suppressed dissipation. As shown in [17], this more complex behavior for the dissipation coefficient with the temperature of the environmental radiation bath makes it possible to produce a large dissipative regimes for WI, in which $Q \gg 1$ can be achieved in this model, with perturbations still consistent with the CMB measurements. On the other hand, the dissipation coefficient in the previous model, Equation (49), in general can only produce perturbations consistent with the CMB observations in the weak dissipative regime of WI, i.e., $Q \ll 1$ (see, e.g., [35]). As this version of the WLI allows WI to be realized in the strong dissipative regime, the cosmological phenomenology that it allows is much richer and has quite appealing features, as we will see in the next section, Section 5.

*3.5. The Axion-like Warm Inflation Model*

A Goldstone boson $\phi$ enjoys a shift symmetry, with only derivatives of $\phi$ appearing in the action. This symmetry can still be softly broken such as to give $\phi$ an ultraviolet (UV) potential and the Goldstone boson becoming a PNGB, like in axion-like models. However, even with the soft breaking of the shift symmetry, radiative and thermal corrections to the axion potential are naturally suppressed by the symmetry properties. This is how axions, for instance, can have very small masses, but that are still protected from large quantum corrections (for a general review on axions and their properties, see, e.g., [71]). Thus, it is natural to try to construct WI by taking the inflaton as an axion-like field. One such successful construction was given in [18]. The relevant interaction of the inflaton $\phi$ in this case is as in an axion interacting with a Yang–Mills field $A_\mu^a$ and with Lagrangian density given by

$$\mathcal{L}_{\text{int}} = \frac{\alpha_g}{8\pi} \frac{\phi}{f} \tilde{F}^{a\,\mu\nu} F_{\mu\nu}^a, \tag{54}$$

where $\tilde{F}^{a\,\mu\nu}$ is the dual gauge field strength, $\tilde{F}^{a\,\mu\nu} = \frac{1}{2}\epsilon^{\mu\nu\alpha\beta}F_{\alpha\beta}^a$, $F_{\mu\nu}^a = \partial_\mu A_\nu^a - \partial_\nu A_\mu^a + gC^{abc}A_\mu^b A_\nu^c$, with $g$ the Yang–Mills coupling and $C^{abc}$ is the structure constant of the non-Abelian group. The coupling constant $\alpha_g$ is $\alpha_g \equiv g^2/(4\pi)$ with $f$ denoting a scale analogous to the axion decay constant in axion-like models [71].

The interaction term Equation (54) leads to a dissipation coefficient that has been shown to be related to the Chern–Simons diffusion rate [72,73] and given by

$$\Upsilon = C_\Upsilon \frac{T^3}{f^2}, \quad C_\Upsilon = \kappa(\alpha_g, N_c, N_f)\alpha_g^5, \tag{55}$$

where $N_c$ is the dimension of the gauge group, $N_f$ is the representation of the fermions if any, and $\kappa$ is a dimensional quantity depending on $N_c$, $N_f$, and $\alpha_g$.

Successful WI dynamics have been shown to be possible in this model [18,74–76]. These studies have also shown the generality of WI in this type of model. Even by starting with quantum initial conditions for the inflaton, e.g., like in CI, the dissipative effects naturally drive the production of a radiation bath and that can thermalize during inflation in these axion-type of models. Hence, a WI dynamics naturally emerges given appropriate parameters in the model [75,76]. Besides of these attractive features, like in the scalar field variant of the WLI model, the dissipation coefficient Equation (55) has the appeal of leading to WI in the strong dissipative regime $Q \gg 1$, yet, still being fully consistent with the CMB perturbations for many different potentials (for a recent exposition on this and a detailed computation of relevant observable quantities, see, e.g., [77]). This will be explicitly shown in the next section. Moreover, it has a minimum setting of parameters and field ingredients that are necessary to lead to WI. This makes this construction specially attractive from the point of view of model building in WI, as far as its simplicity is concerned.

## 4. Cosmological Perturbations in WI

As we discussed previously in Section 2, when $T > H$, thermal fluctuations of the inflaton field will become dominant. As a consequence of this, the source of density fluctuations in the WI scenario is the thermal fluctuations in the radiation field, which are

then transferred to the inflaton field as adiabatic curvature perturbations. This is much different than in the CI case, where quantum fluctuations are responsible for generating the seeds for large-scale structure formation. We will discuss here how dissipative effects change the dynamics of inhomogeneous fluctuations of the inflaton field. To study the scalar perturbations for WI, we start with the fully perturbed FLRW metric (here, we are including only scalar perturbations), which is given by

$$ds^2 = -(1 + 2\alpha)dt^2 - 2a\partial_i\beta dx^i dt + a^2\left[\delta_{ij}(1 + 2\varphi) + 2\partial_i\partial_j\gamma\right]dx^i dx^j, \tag{56}$$

where $\alpha$, $\beta$, $\gamma$, and $\varphi$ are the spacetime-dependent perturbed-order variables. Moreover, one needs to expand the inflaton field, the radiation energy density, and also the radiation pressure around their background values in the FRLW metric, such that

$$\phi(\mathbf{x}, t) = \bar{\phi}(t) + \delta\phi(\mathbf{x}, t), \tag{57}$$
$$\rho_r(\mathbf{x}, t) = \bar{\rho}_r(t) + \delta\rho_r(\mathbf{x}, t), \tag{58}$$
$$p_r(\mathbf{x}, t) = \bar{p}_r(t) + \delta p_r(\mathbf{x}, t), \tag{59}$$

where $\bar{\phi}(t)$, $\bar{\rho}_r(t)$, and $\bar{p}_r(t)$ are the background values for the inflaton field, the radiation energy density, and the pressure, while $\delta\phi(x, t)$, $\delta\rho_r(\mathbf{x}, t)$, and $\delta p_r(\mathbf{x}, t)$ are, respectively, their corresponding perturbations. Since dissipation coefficient is generally a function of $\phi$ and $T$, Equation (16), it must be treated similarly, i.e., $Y(\mathbf{x}, t) = \bar{Y}(t) + \delta Y(\mathbf{x}, t)$. Hence, working in momentum space, defining the Fourier transform with respect to the co-moving coordinates, the equation of motion for the radiation and momentum fluctuations with co-moving wavenumber $k$ are found to be given by [42]:

$$\delta\dot{\rho}_r + 3(1 + \omega_r)H\delta\rho_r = (1 + \omega_r)\rho_r(\kappa - 3H\alpha) + \frac{k^2}{a^2}\Psi_r + \delta Q_r + Q_r\alpha, \tag{60}$$

$$\dot{\Psi}_r + 3H\Psi_r = -\omega_r\delta\rho_r - (1 + \omega_r)\rho_r\alpha + J_r, \tag{61}$$

where $\Psi_r$ is the momentum perturbation, $\omega_r$ is the equation of state for the radiation fluid, $\chi = a(\beta + a\dot{\gamma})$, $\kappa = 3(H\alpha - \dot{\phi}) + k^2\chi/a^2$, and $J_r = -Y\dot{\phi}\delta\phi$ is the momentum source. Furthermore, $Q_r = Y\dot{\phi}^2$ and its perturbation $\delta Q_r$ is given by

$$\delta Q_r = \delta Y\dot{\phi}^2 + 2Y\dot{\phi}\delta\dot{\phi} - 2\alpha Y\dot{\phi}^2. \tag{62}$$

In addition to Equations (60) and (61), one also has the evolution equation for the inflaton field fluctuations $\delta\phi$. Assuming the universe remains near thermal equilibrium during WI, the fluctuations of the inflaton field obey a fluctuation–dissipation relation [52]. Therefore, the evolution of the inflaton fluctuations is achieved by perturbing the inflaton field equation, and adding stochastic quantum and thermal white noise terms, following the fluctuation–dissipation theorem, as follows (for further details, see also [78]):

$$\delta\ddot{\phi} + 3H\delta\dot{\phi} + \left(\frac{k^2}{a^2} + V_{,\phi\phi}\right)\delta\phi$$
$$= \xi_q + \xi_T - \delta Y\dot{\phi} + \dot{\phi}(\kappa + \dot{\alpha}) + (2\ddot{\phi} + 3H\dot{\phi})\alpha - Y(\delta\dot{\phi} - \alpha\dot{\phi}), \tag{63}$$

where $\xi_{q,T}$ are stochastic Gaussian sources related to the quantum and thermal fluctuations with zero mean, $\langle\xi_T\rangle = \langle\xi_q\rangle = 0$, and with appropriate amplitudes, which are defined by the two-point correlation functions:

$$\langle\xi_T(\mathbf{k}, t)\xi_T(\mathbf{k}', t')\rangle = \frac{2YT}{a^3}\delta(t - t')(2\pi)^3\delta(\mathbf{k} + \mathbf{k}'), \tag{64}$$

$$\langle\xi_q(\mathbf{k}, t)\xi_q(\mathbf{k}', t')\rangle = \frac{H^2(9 + 12\pi Q)^{1/2}(1 + 2n_*)}{\pi a^3}\delta(t - t')(2\pi)^3\delta(\mathbf{k} + \mathbf{k}'), \tag{65}$$

where $n$ denotes the inflaton statistical distribution due to the presence of the radiation bath. For a thermal equilibrium distribution[3], it assumes the Bose–Einstein distribution form, i.e., $n = 1/[\exp(H/T) - 1]$. To find the complete evolution of perturbations, one needs to specify the fluctuation of the dissipation coefficient. Considering the parameterization defined by Equation (16), $\delta Y$ can be written as

$$\delta Y = Y\left[c\frac{\delta T}{T} + p\frac{\delta\phi}{\phi}\right]. \tag{66}$$

Although dissipation implies departures from thermal equilibrium in the radiation fluid, the system has to be close-to-equilibrium for the calculation of the dissipative coefficient to hold, therefore, $p_r = \rho_r/3$, $\rho_r \propto T^4$ and $\delta T/T = \delta\rho_r/(4\rho_r)$. Hence, the $\delta Q_r$ term in Equation (60) can be explicitly written as

$$\delta Q_r = 3HQ\dot\phi^2\left(\frac{c\delta\rho_r}{4\rho_r} + \frac{p\delta\phi}{\phi}\right) + 6HQ\dot\phi\delta\dot\phi - 6HQ\dot\phi^2\alpha. \tag{67}$$

From the above relations, the complete system of first-order perturbation equations for WI are given by

$$\delta\ddot\phi = -3H(1+Q)\delta\dot\phi - \left(\frac{k^2}{a^2} + V_{,\phi\phi} + \frac{3pHQ\dot\phi}{\phi}\right)\delta\phi + \xi_q + \xi_T - \frac{cH}{\dot\phi}\delta\rho_r + \dot\phi(\kappa + \dot\alpha)$$
$$+ [2\ddot\phi + 3H(1+Q)\dot\phi]\alpha, \tag{68}$$
$$\delta\dot\rho_r = -H\left(4 - \frac{3cQ\dot\phi^2}{4\rho_r}\right)\delta\rho_r + \frac{k^2}{a^2}\Psi_r + 6HQ\dot\phi\delta\dot\phi + \frac{3pHQ\dot\phi^2}{\phi}\delta\phi + \frac{4\rho_r}{3}\kappa$$
$$- 3H\left(Q\dot\phi^2 + \frac{4\rho_r}{3}\right)\alpha, \tag{69}$$
$$\dot\Psi_r = -3H\Psi_r - 3HQ\dot\phi\delta\phi - \frac{1}{3}\delta\rho_r - 4\rho_r\frac{\alpha}{3}. \tag{70}$$

One can immediately realize from Equations (68) and (69) that the inflaton fluctuations $\delta\phi$ are coupled to the radiation fluctuations $\delta\rho_r$ when $c \neq 0$. As first realized in [80], such coupling results in a growing mode, if $c > 0$ or a decreasing mode for $c < 0$ in the curvature power spectrum as the dissipation ratio $Q$ increases. In other words, dissipation will increase the temperature more in regions where it is already higher than average making the power spectrum scale-dependent and blue (red) for $c > 0$ ($c < 0$). The effect of this coupling between inflaton and radiation perturbations in WI is typically modeled by a function $G(Q)$, as we will show below.

The sets of perturbation Equations (68)–(70) are gauge-ready equations. Hereafter, one can rewrite these equations in terms of gauge-invariant quantities (see [42]) or use an appropriate gauge choice [81,82]. Although any appropriate gauge can be chosen, here we will make use of the zero-shear gauge, i.e., $\chi = 0$. This in particular has been shown to be more advantageous, as far as numerical stability is concerned when solving the complete system of equations. Thus, in the zero-shear gauge, the relevant metric equations become:

$$\kappa = \frac{3}{2M_{\text{Pl}}^2}(\dot\phi\delta\phi - \Psi_r), \tag{71}$$

$$\alpha = -\varphi, \tag{72}$$

$$\dot\varphi = -H\varphi - \frac{1}{3}\kappa. \tag{73}$$

Once all the relevant perturbation equations have been defined, the power spectrum is determined from the definition of the co-moving curvature perturbation [83]:

$$\Delta_{\mathcal{R}}(k) = \frac{k^3}{2\pi^2} \left\langle |\mathcal{R}|^2 \right\rangle, \tag{74}$$

where $\langle \ldots \rangle$ means here the ensemble average over different realizations of the noise terms in Equation (68) and which satisfy Equations (64) and (65). Finally, the general expression for $\mathcal{R}$ is composed of contributions from the inflaton momentum perturbations and from the radiation momentum perturbations [42]:

$$\mathcal{R} = \sum_{i=\phi,r} \frac{\bar{\rho}_i + \bar{p}_i}{\bar{\rho} + \bar{p}} \mathcal{R}_i, \tag{75}$$

$$\mathcal{R}_i = -\varphi - \frac{H}{\bar{\rho}_i + \bar{p}_i} \Psi_i, \tag{76}$$

with $\bar{p} = \bar{p}_\phi + \bar{p}_r$, $\bar{\rho}_\phi + \bar{p}_\phi = \dot{\phi}^2$ and $\bar{\rho}_r + \bar{p}_r = 4\bar{\rho}_r/3 = Q\dot{\phi}^2$. As we have already mentioned, the inflaton field fluctuations and the radiation field fluctuations are coupled together in the case when $c \neq 0$. Therefore, to obtain the power spectrum for the inflaton fluctuations, one typically needs to numerically solve Equations (68)–(70) along with the appropriate set of metric perturbation equations. However, an explicit analytic expression for the scalar of curvature power spectrum can be obtained for dissipation coefficients which are independent of temperature, i.e., when $c = 0$. In this case, the equations for the inflaton and the radiation fluctuations become decoupled and one can obtain the curvature power spectrum using Green function techniques (see [78,80] for details). This leads to the explicit result for the curvature perturbation,

$$\Delta_{\mathcal{R}}\big|_{c=0} = \frac{H_*^3 T_*}{4\pi^2 \dot{\phi}_*^2} \left[ \frac{3Q_*}{2\sqrt{\pi}} 2^{2\alpha} \frac{\Gamma(\alpha)^2 \Gamma(\nu-1)\Gamma(\alpha-\nu+3/2)}{\Gamma\left(\nu-\frac{1}{2}\right)\Gamma(\alpha+\nu-1/2)} + \frac{H_*}{T_*} \coth\left(\frac{H_*}{2T_*}\right) \right], \tag{77}$$

where $\nu = 3(1+Q)/2$, $\alpha = \sqrt{\nu^2 + 3\beta_V Q/(1+Q) - 3\eta_V}$, with $\beta_V$ and $\eta_V$ as already defined before, e.g., below Equation (12). In Equation (77), we also have that $\Gamma(x)$ is the Gamma-function, while the subindex $*$ means that all quantities are to be evaluated at the Hubble crossing time, e.g., when $k_* = a_* H_*$. This specific point during inflation and where the relevant modes crosses the Hubble radius can be defined as follows. First we note that we can relate the mode with co-moving wavenumber $k_*$ that crossed the Hubble horizon, $a_* H_* = k_*$, with the one at present time, $a_0 H_0$, as [84]

$$\frac{k_*}{a_0 H_0} = \frac{a_*}{a_{\text{end}}} \frac{a_{\text{end}}}{a_{\text{reh}}} \frac{a_{\text{reh}}}{a_0} \frac{H_*}{H_0}, \tag{78}$$

where $a_*/a_{\text{end}} = \exp(-N_*)$ and $N_*$ is the number of e-folds lasting from the point where the modes left the Hubble radius until the end of inflation, where the scale factor is $a_{\text{end}}$. Now, as we have already seen in Section 2, WI ends when the radiation energy density takes over and starts dominating. Since in WI there is no need for a reheating phase after the end of inflation, this removes a significant source of uncertainty that is present in CI models, which is related the specific duration of the reheating phase and which affects the predictions that (cold) inflation can make regarding observables, such as, for instance, the tensor-to-scalar ratio and the spectral tilt of the spectrum. Hence, in WI, we can simply set $a_{\text{end}}/a_{\text{reh}} = 1$ in Equation (78). This uniquely specifies the relevant number of e-folds $N_*$ in WI, which from Equation (78), can be shown to be obtained through the relation [27]:

$$\frac{k_*}{a_0 H_0} = e^{-N_*} \left[ \frac{43}{11 g_s(T_{\text{end}})} \right]^{1/3} \frac{T_0}{T_{\text{end}}} \frac{H_*}{H_0}, \tag{79}$$

where $T_{\text{end}}$ is the temperature at the end of WI and $g_s(T_{\text{end}})$ is the number of relativistic degrees of freedom at that temperature.

Thus, by returning again to Equation (77) and dropping the slow-roll coefficients as a first-order approximation, we have that $\alpha \simeq \nu$ and Equation (77) can be very well approximated by the result[4]:

$$\Delta_{\mathcal{R}}\big|_{c=0} \simeq \left(\frac{H_*}{\dot{\phi}_*}\right)^2 \left(\frac{H_*}{2\pi}\right)^2 \left(1 + 2n_* + \frac{2\sqrt{3}\pi Q_*}{\sqrt{3 + 4\pi Q_*}}\frac{T_*}{H_*}\right). \tag{80}$$

In the case of dissipation coefficients that have an explicit temperature dependence, one needs to solve Equations (68)–(70) to fully determine what is the effect of the coupling to radiation on the inflaton power spectrum. This then leads to a change of the result given by Equation (80), which is usually parameterized by a multiplicative function of the dissipation coefficient, $G(Q)$, in Equation (80), such that

$$\Delta_{\mathcal{R}}\big|_{c\neq 0} = \Delta_{\mathcal{R}}\big|_{c=0} G(Q_*). \tag{81}$$

Neglecting both the metric perturbations and the slow-roll parameters and also the field dependence in Equation (66), in [80], it was found that $G(Q)$ has an asymptotic behavior for $Q \gg 1$ given by $G(Q) \sim (Q/Q_c)^{3c}$, where $Q_c$ is a constant depending on the value of $c$. Later, in [42], it was shown that accounting for the neglected slow-roll first-order quantities as considered in [80] could actually overestimate the growing function $G(Q)$ by many orders of magnitude at large $Q$ values[5]. For now, we only know how to obtain the growing function $G(Q)$ by numerically solving the full set of coupled background and perturbation equations and then numerically fitting the spectrum for a given form of the dissipation coefficient. For some specific representative cases of dissipation coefficients, for instance, for those with powers in the temperature $c = 3$ (cubic), $c = 1$ (linear), and $c = -1$ (inverse), the function $G(Q)$ for not too large values of $Q$, $Q_* \lesssim 200$, can be well approximated by[6]:

$$G_{\text{cubic}}(Q_*) \simeq 1 + 4.981 Q_*^{1.946} + 0.127 Q_*^{4.330}, \tag{82}$$

$$G_{\text{linear}}(Q_*) \simeq 1 + 0.335 Q_*^{1.364} + 0.0185 Q_*^{2.315}, \tag{83}$$

$$G_{\text{inverse}}(Q_*) \simeq \frac{1 + 0.4 Q_*^{0.77}}{(1 + 0.15 Q_*^{1.09})^2}. \tag{84}$$

Once the growing function $G(Q)$ is specified, then from the expression for the power spectrum in WI, we can, for instance, obtain the scalar spectral index $n_s$, which is defined in general as

$$n_s - 1 \equiv \lim_{k \to k_*} \frac{d\ln\Delta_{\mathcal{R}}}{d\ln k} \simeq \lim_{k \to k_*} \frac{d\ln\Delta_{\mathcal{R}}}{dN}. \tag{85}$$

Using that

$$\frac{d\ln k}{dN} \approx 1 - \epsilon_V/(1+Q), \tag{86}$$

we obtain, for instance, that [77]

$$\begin{aligned}
n_s &= 1 + \frac{(1+Q)}{1+Q-\epsilon_V}\left[\frac{d\ln\left(\frac{T}{H}\right)}{dN}\right. \\
&\quad + \frac{\frac{d\ln Q}{dN}\left(-3 + Q\{3 + 2\pi[-1 + Q(3 + \sqrt{9 + 12Q\pi})]\}\right)}{(1+Q)\left[3 + Q\pi(4 + \sqrt{9 + 12Q\pi})\right]} \\
&\quad + \left.\frac{d\ln Q}{dN}\mathcal{A}(Q) + \frac{-6\epsilon_V + 2\eta_V}{1+Q}\right],
\end{aligned} \tag{87}$$

where the function $\mathcal{A}(Q)$ is given by:

$$\mathcal{A}(Q) = \frac{3 + 2\pi Q}{3 + 4\pi Q} + Q \frac{d \ln G(Q)}{dQ},$$
(88)

and the derivatives with respect to the number of e-folds appearing in Equation (87) are given by Equations (18) and (19). One can easily observe from Equation (87) that it reduces to the spectral index for CI, i.e., $n_s - 1 = -6\epsilon_V + 2\eta_V$, when $Q \to 0$ and $T \to 0$.

Precise expressions for the spectral tilt and also for the running, $\alpha_s = dn_s(k)/d\ln(k)$, and for the running of the running, $\beta_s = d\alpha_s(k)/d\ln(k)$, have been given in [77] and which are specially useful to analyze WI in the strong dissipative regime.

While the curvature power spectrum is significantly modified by dissipative effects, the tensor power spectrum remains unaffected since the gravitational interaction is weak[7]. Therefore, we can write the tensor power spectrum as in CI,

$$\Delta_T = \frac{8}{M_{\rm Pl}^2} \left( \frac{H_*}{2\pi} \right)^2.$$
(89)

From Equation (89), one can define the tensor spectral index similar to the scalar spectral one,

$$n_t \equiv \lim_{k \to k_*} \frac{d \ln \Delta_T}{d \ln k} \simeq -2\epsilon_V.$$
(90)

Finally, the tensor-to-scalar ratio $r$ is given by:

$$r = \frac{\Delta_T}{\Delta_{\mathcal{R}}} = \frac{16\epsilon_V}{1 + Q} \mathcal{F}^{-1},$$
(91)

from which one can see that the CI result gets suppressed in WI by the factors $1 + Q$ and $\mathcal{F}$, where

$$\mathcal{F} = \left( 1 + 2n_* + \frac{2\sqrt{3}\pi Q_*}{\sqrt{3 + 4\pi Q_*}} \frac{T_*}{H_*} \right) G(Q_*).$$
(92)

In particular, as a consequence of this, WI can produce a much reduced tensor-to-scalar ratio as compared to CI for the same type of primordial inflaton potentials. In particular, well-motivated potentials that became excluded in CI, such as the simple quadratic and quartic power law inflaton potentials, can be made perfectly consistent with the CMB data, e.g., from Planck (see, for instance [22,35]). Equation (91) also implies that the Lyth bound [83] found in the context of CI and which relates the inflaton field excursion to the tensor-to-scalar ratio, $\Delta\phi/M_{\rm Pl} \sim \sqrt{r}$, will be violated in the WI scenario. Similarly, the dissipative effects will violate the consistency relation of CI where $r = -8n_t$. The violation of this relation happens in WI even in the weak dissipative regime, $Q < 1$, since the fluctuations are in a thermally excited state for $T > H$, i.e., $n_* \neq 0$. We will see below, in Section 5, that such smoking gun features, together with the much richer dynamics allowed by WI, can make WI to satisfy the swampland conjectures inspired from string theory.

It is useful to compare the typical energy scales from WI with those from CI[8]. We note that the amplitude of gravitational wave perturbations, as well as the potential energy scale of CI, is constrained by the CMB. In particular, in CI, we can relate the energy scale of the inflaton potential $V_*$ directly in terms of the tensor-to-scalar ratio as

$$V_*^{\rm CI} = \frac{3\pi^2 \Delta_{\mathcal{R}}}{2} r M_{\rm Pl}^4 < \left( 1.4 \times 10^{16} {\rm GeV} \right)^4,$$
(93)

where we have used Equation (89), with the value for the amplitude of the scalar power spectrum [93], $\ln(10^{10}\Delta_{\mathcal{R}}) \simeq 3.047$ (TT,TE,EE-lowE+lensing+BAO 68% limits), i.e.,

$\Delta_{\mathcal{R}} \simeq 2.1 \times 10^{-9}$, and that $r < 0.036$, from the combined BICEP/Keck and Planck results [94]. In WI, from Equation (91), we have that

$$\Delta_T = \frac{16\epsilon_V^{wi}}{1+Q}\mathcal{F}^{-1}\Delta_R. \tag{94}$$

Using once again Equation (89) and the first Friedmann equation in the slow-roll limit, we can find the energy scale in the case of WI (at pivot scale $k_*$) as

$$V_*^{\mathrm{WI}} = \frac{3\pi^2}{2}M_{\mathrm{Pl}}^4\frac{16\epsilon^{ci}}{1+Q_*}\mathcal{F}^{-1}\Delta_R. \tag{95}$$

This can be compared with the equivalent result in CI[9],

$$V_*^{\mathrm{CI}} = \frac{3\pi^2}{2}M_{\mathrm{Pl}}^4 16\epsilon_V^{ci}\Delta_R. \tag{96}$$

Thus,

$$\frac{V_*^{\mathrm{WI}}}{V_*^{\mathrm{CI}}} = \mathcal{F}^{-1}, \tag{97}$$

which means that the energy scale for the inflaton in WI is always smaller than the one expected from CI, Equation (93). Likewise, we have for the ratio between Hubble scales in WI and in CI is given by:

$$\frac{H_*^{\mathrm{WI}}}{H_*^{\mathrm{CI}}} = \frac{1}{\sqrt{\mathcal{F}}}. \tag{98}$$

As WI tends in general to predict a smaller tensor-to-scalar ratio than that of CI, this then translates in smaller energy scales in WI than in CI.

To conclude this section, let us briefly discuss the amount of non-Gaussianity that is produced by dissipative effects during WI. The first rigorous attempt to estimate the nonlinearity effects by WI was done in [95]. In that reference, by considering the dissipation coefficient just as a function of the inflaton field ($c = 0$), it was found that the nonlinearity parameter could be approximated as $f_{NL}^{\mathrm{warmS}} = -15\ln(1 + Q/14) - 5/2$ in the strong dissipative regime. Then, the results were later generalized in [96] for the case of temperature-dependent dissipation coefficients ($c \neq 0$), while still in the strong dissipative regime. It was also realized that the coupling of inflaton and the radiation field fluctuations due to the temperature-dependent dissipation coefficient would potentially make the nonlinearity effects stronger. In fact, it was found that the nonlinearity parameter will be larger for larger values for the exponent $c$, the power of the temperature in the dissipation coefficient. However, the most concrete results were obtained in [97]. Through a numerical method developed in [97], the nonlinearity parameter was calculated for the general form of the dissipation coefficient and in both weak and strong dissipative regimes. The results obtained for the nonlinearity parameter turned out to be smaller than what was found originally in [95] for the strong dissipative regime. According to the results reported in [97], the non-Gaussianity parameter can significantly depend on the values of $T/H$ and also $Q$ in the weak dissipation regime, i.e., for $Q < 0.1$ and it reaches its maximum at $Q \sim 10^{-3}$, when the thermal fluctuations start to become dominant. However, when the dissipation ratio is large, $Q > 1$, the nonlinearity parameter is independent of $T/H$ and it only mildly depends on $Q$. This happens because of the existence of the growing mode in the power spectrum, which enhances the amplitude of the power spectrum by a factor $Q^\alpha$ and modifies the bispectrum by a factor $Q^{2\alpha}$, while the effects partially cancel out in the $f_{NL}$. Furthermore, the nonlinearity parameter is larger for large value of $c$ in the weak dissipation regime. However, as the dissipation ratio increases, it results in comparable

nonlinearity parameters in the strong dissipative regime. Apart from the magnitude of the nonlinearity parameter, it was also found in [97] that WI predicts two distinct shapes for the nonlinearity, depending whether WI happens in the weak or in the strong dissipative regime. This in particular makes WI distinguishable from CI if non-Gaussianities are detected in the near future[10].

## 5. Swampland Criteria, Observational Constraints, and Other Applications

In this section, we briefly review the implications of the swampland conjectures that have been discussed recently and put forward when applied to inflation models. We discuss how the swampland constraints are indicative of ruling out most, if not all, single field CI models, while WI is able to be in accordance with these swampland constraints and, at the same time, also satisfy all current observational constraints. Finally, we conclude the section by discussing some of the recent applications concerning WI and the efforts to using it to elucidate some of the outstanding pre/post-inflationary phenomena.

### 5.1. WI and Swampland Conjectures

It is always assumed that inflation can be described using low energy effective field theory (EFT), since the energy scale of inflation is below the Planck scale when the currently observable scale exits the Hubble horizon. However, this does not mean that any inflationary model can be ultraviolet-complete. Therefore, it is important to try to distinguish those EFTs that can be consistently embedded into a quantum theory of gravity from those that cannot. This was the starting point of the swampland program and aiming at distinguishing those EFTs, which belong to the landscape of string theory, from those that inhibit in the swampland. By having the correct criteria to identify the boundary between the landscape and the swampland, this resulted in a series of conjectures known as the *swampland conjectures* (see, e.g., [99], for a review of these ideas). In fact, these conjectures, although speculative, are the very first theoretical constraints originating from string theory that can have direct implications on the inflationary cosmology.

As discussed in [100], low-energy EFTs can become inapplicable during compactification in string theory. This is because the mass of quantum gravity states decreases exponentially rapid as the field excursion in the moduli space increases, i.e., a tower of massive states becomes exponentially light as $\exp(-\alpha \Delta\phi / M_{\rm Pl})$, with $\alpha \sim \mathcal{O}(1)$. As a result, if the scalar field has excursions beyond the Planck scale, $\Delta\phi > M_{\rm Pl}$, a large number of new light states must be considered. Thus, it has then been speculated that the change of any scalar field arising in the EFT in its field space is confined by a positive constant number of order one,

$$\frac{\Delta\phi}{M_{\rm Pl}} < c_1. \tag{99}$$

The above constraint (99) is called the *swampland distance conjecture (SDC)*. This bound arises from the fact that if the scalar field moves by a range that is larger than the upper bound given by Equation (99), thus making a super-Planckian field excursion, then a tower of new string states becomes light, as said above, and they must be included in the low-energy EFT. The SDC has an immediate implication for inflation models. In fact, SDC indicates that large field inflationary models, which generically are known to lead to super-Planckian field excursions in the context of CI (we will see below how WI can evade the constraint given by Equation (99)), tend to be ruled out as a consistent EFT model. Since the field excursion is related to the tensor-to-scalar ratio through the Lyth bound, hence, one may infer that SDC is indicating that the Lyth bound should be violated in a non-trivial way such as to make inflation consistent with SDC and the observational data. As we have already discussed in the previous Section 4, WI naturally violates the Lyth bound. Furthermore, in WI the inflaton field can be slowed down not only because of the Hubble friction but also from the explicit dissipation term. If the dissipation term is large enough, in particular in the strong dissipative regime $Q \gg 1$, the inflaton excursions can

be made sufficiently short such as to satisfy Equation (99). This feature of WI in the strong dissipative regime has been observed by many recent realizations of this regime (see, for instance, [25,27,88]).

Several years after introducing the SDC, it was discussed that constructing a metastable de Sitter vacuum is notoriously difficult in string theory. This has lead to the conjecture that metastable de Sitter vacua should belong to the swampland rather than to the landscape of string theory. As a consequence of this difficulty in building appropriate de Sitter vacua, it can then be translated in conditions that potentials for scalar fields should satisfy and which are known as the swampland *de Sitter conjecture* (SdSC). These conditions imply in bounds on the slope of the scalar potentials in an EFT and which can be expressed in the form [37,101],

$$\frac{|\nabla V|}{V} \geq \frac{c_2}{M_{\text{Pl}}}, \tag{100}$$

or

$$\frac{min(\nabla_i \nabla_j V)}{V} \leq -\frac{c_3}{M_{\text{Pl}}^2}, \tag{101}$$

where $\nabla$ is the gradient in field space, $c_2$ and $c_3$ are universal and positive constants of order one, and $min(\nabla_i \nabla_j V)$ is the minimum eigenvalue of the Hessian $\nabla_i \nabla_j V$ in an orthonormal frame. The conditions given by Equations (100) and (101) mean that either a useful potential has to be sufficiently steep, or else sufficiently tachyonic near its maximum, if it has one. It was shown in [37] that the first de Sitter condition, given by Equation (100), is related to the distance condition, Equation (99), in the weak coupling regime and which can be demonstrated by using Bousso's covariant entropy bound [102]. Looking at the conditions given by Equations (100) and (101), one can easily realize that these conditions can be translated into conditions on the slow-roll coefficients $\epsilon_V$ and $\eta_V$, such that they must satisfy [101]:

$$\epsilon_V \equiv \frac{M_{\text{Pl}}^2}{2} \left( \frac{V_{,\phi}}{V} \right)^2 \geq \frac{c_2^2}{2}, \quad \text{or} \quad \eta_V \equiv M_{\text{Pl}}^2 \frac{V_{,\phi\phi}}{V} \leq -c_3, \tag{102}$$

thus, requiring the potential slow-roll parameters to be of order unity. This is obviously in contrast to the slow-roll conditions imposed in CI models, where $\epsilon_V \ll 1$ and $\eta_V \ll 1$. Hence, the first condition rules out all single field CI models, while the second condition is still consistent with those inflationary potential with tachyonic instability, such as hilltop potentials [103]. Recalling that $\epsilon_H = -\dot{H}/H^2$, and that in the context of CI $\epsilon_H \simeq \epsilon_V$, hence, in the CI scenario one cannot achieve slow-roll inflationary phase for steep potentials. Moreover, the SdSC also excludes any extrema of scalar field potentials in field space, i.e., $|\nabla_\phi V|/V \to 0$. This is in clear contrast with the reheating phase after the end of inflation, which requires the inflaton potential to have a minimum. Therefore, the SdSC also indicates that the inflaton should terminate using a new mechanism rather than the so-called reheating phase in the CI scenario. Both of these issues are again possible to be overcome in the context of WI. Recalling again that in WI both slow-roll parameters $\epsilon_V$ and $\eta_V$ are replaced by $\epsilon_H \simeq \epsilon_{wi} = \epsilon_V/(1+Q)$ (see, e.g., Equation (11)) and $\eta_{wi} = \eta_V/(1+Q)$. Hence, in WI, there is no problem of having $\epsilon_V$ and $\eta_V$ larger than 1, provided that $Q \gg 1$. WI in the strong dissipative regime comes again to the rescue, making the inflaton perfectly consistent with the SdSC as shown explicitly in the models worked out in [25,27,88], for example.

It is also important to note that the so-called $\eta$-problem of cold inflation [104] is reminiscent of the SdSC. The $\eta$-problem is related to the fact that to have inflation, one requires $V_{\phi\phi} \equiv m_\phi^2 \ll H^2$, such that Hubble friction dominates during inflation. However, the inflaton potential is prone to receive large corrections and which can drive the inflaton mass above the Hubble scale $H$. This is the "eta-problem", which appears in, e.g., F-term

supergravity and string theory. If the inflaton mass is driven to super-Hubble values, we have a tension with the slow-roll conditions. As argued a long time ago already [5,19,20], WI in the strong dissipative regime provides a natural solution for the $\eta$-problem, as we have seen also here when the SdSC constraint is satisfied by WI.

As WI intrinsically leads to entropy production as a consequence of particle production, it might change the above swampland conditions. This is specially for the case of the de Sitter conjecture, which, as mentioned above, can be related to entropic phenomena. It has been shown in [26] that extra entropy produced as a result of particle production in the context of WI has a negligible effect and WI remains fully consistent with the SDC and SdSC.

In addition to the above swampland conjectures, very recently, another conjecture called the *Trans-Planckian censorship Conjecture* (TCC) has also been proposed [39,41]. The TCC proposal is based on the trans-Planckian problem of inflationary models, which severely constrain inflationary models. The TCC requests that the Hubble horizon must hide sub-Planckian modes during the early stages of accelerated expansion,

$$\left(\frac{a_f}{a_i}\right)\ell_{\text{Pl}} < \frac{1}{H_f}, \tag{103}$$

where $a_i$ and $a_f$ are, respectively, the scale factors at the beginning and at the end of the evolution, $H_f$ is the Hubble parameter at the end of that evolution, and $\ell_{\text{Pl}}$ is a length scale of the order of the Planck scale[11]. The TCC bound can be translated into an upper bound on the duration of inflation. If we assume a constant Hubble horizon, or a period of quasi-de Sitter inflation and instant reheating phase at the end of inflation, the TCC implies an upper limit for the energy scale of inflation [41]:

$$V^{\frac{1}{4}} < 6 \times 10^8 \text{GeV} \sim 3 \times 10^{-10} \text{M}_{\text{Pl}}, \tag{104}$$

which, in turn, it can be translated to an upper bound on the tensor-to-scalar ratio parameter,

$$r < 10^{-31}. \tag{105}$$

The result given by Equation (105) tightly constrains the slow-roll epoch needed to resolve the shortcomings of the standard big bang cosmology according to the standard CI picture. In fact, to make CI models compatible with the TCC, one needs to modify the dynamics of the CI scenario in such a way that the tensor-to-scalar ratio can be suppressed significantly. By accounting for a non-standard evolution after the end of inflation, it has been shown [106] that it can lead to a more accessible result for $r$, $r < 10^{-8}$, which is weaker than the upper bound given by Equation (105). However, the result is still very constraining on the energy scale of inflation. Turning now to WI, one can see how it can evade the TCC bound discussed here. As explained in Section 4, the dissipative effects in WI produce a modified primordial scalar of the curvature power spectrum, Equation (80), which is enhanced by the dissipation and thermal effects. Again, for $Q \gg 1$, this results in a highly suppressed tensor-to-scalar ratio, Equation (91), which can make WI also consistent with the TCC [25,27,88]. All of this is achieved with WI also leading to consistent results for the spectral tilt $n_s$ and also for its runnings [77], thus consistent with the current results from Planck [108].

There have been also discussions concerning eternal inflation and the swampland and whether eternal inflation would be in the landscape or in the swamp of EFTs in the context of quantum gravity theories (see, e.g., [103,109,110]). Interestingly, WI, again, here seems to play a significant role. The dissipative and thermal characteristics of WI have been shown to provide a way of suppressing the emergence of eternal inflation and, in the strong dissipative regime, even possibly avoiding it to happen [111].

Although all these swampland conjectures explained above are speculative and there is no strong evidence to strengthen them, it is worth noting that WI is able to satisfy and to be consistent with all of the constraints that these conditions bring, in a natural way.

*5.2. Other Applications and Recent Results in WI*

As we have discussed in Sections 2 and 4, WI enlarges the class of potentials that can be made observationally consistent with the observational data in comparison to CI scenario. For instance, simple potentials for the inflaton, such as the quadratic and quartic monomials potentials, the Higgs-like symmetry breaking potential and the Coleman–Weinberg-type of potentials, which are all well motivated in the context of particle physics, have been shown to the excluded by the Planck data in the context of CI [108]. However, in WI, because of the different dynamics resulting from the dissipative effects, they can all be made fully consistent with the observations and this can happen for a large range of space of parameters, as discussed, e.g., in [35].

Considering also the natural inflaton potential [112], i.e., $V(\phi) = \Lambda^4[1 + \cos(\phi/f)]$ in the WI scenario, it was found that although in CI the model is now borderline-consistent with the observations for super-Planckian values for the decay constant $f$, it can become consistent with the observations even in the case of sub-Planckian values for $f$ in the context of the WI picture, as shown in case of a simplified constant dissipation coefficient form [113–115]. A sub-Planckian value for $f$ is favorable from a model building and EFT perspective. However, very recently [116], this model was again reconsidered in WI in the cases of linear and cubic temperature-dependent dissipation coefficients and it was found that due to the growing mode function $G(Q)$, the model is not consistent with the observations in the case of sub-Planckian values for $f$.

The results and studies in WI can also be extended to the case of non-canonical type of models. For instance, it has been shown that for the quartic potential, $V = \lambda\phi^4$, the combination of dissipative effects with G-inflaton makes this potential consistent with observations even for large values for the self-coupling $\lambda$ [89].

Moreover, there are several papers in the literature which studied WI using some quasi-de Sitter form of scale factor, e.g., intermediate and logamediate inflation, and other non-canonical versions of field theory, e.g., tachyonic fields, in the high dissipative regime [117–125], while there have been also many studies involving different forms of inflaton potentials and in other interesting contexts [79,111,126–144].

The analysis of the dynamics from the point of view of a dynamical system realization is important for many reasons. It not only can bring general information about the dynamics that the system can present but also regarding its stability. In this context, several studies have analyzed WI from a dynamical system perspective, as, for example, in [43–45,145–147]. WI has also been contrasted with the observational data through explicit statistical analysis, as in [35,148–151]. The inflaton itself in WI can be a source and responsible for cosmic magnetic field generation [152], and in combination with the intrinsic dissipative effects lead to a successful baryogenesis scenario [153–155]. Moreover, the gravitino problem was also considered in the context of WI [156,157]. More recently, a focus has been given to WI in topics such as dark energy [158–163], dark matter [70,164–167], and in problems related to primoridial black hole formation and in providing additional sources of gravitational waves [168–172], just to cite a few of the many recent developments in the context of WI.

## 6. Conclusions

Warm inflation provides a framework for understanding the dynamics of the early universe from a perspective that differs from that of the standard cold inflation. Warm inflation accounts for the explicit dissipative effects that are expected to be present in any dynamical physical theory with interacting particles and fields. When these dissipative effects are strong enough to overcome the dilution of the radiation during the rapid expansion in the primordial inflation epoch, a rich dynamics and new phenomena emerge that are not present during the usual cold inflation picture.

In warm inflation, the generation of density perturbations can be entirely classical, formed by thermal fluctuations instead of quantum fluctuations in the case of cold inflation. This overcomes several issues related to classicalization of perturbations in the case of cold inflation. Furthermore, warm inflation predicts a very small, if not substantially small, tensor-to-scalar ratio due to the thermal and dissipative effects. As a result, simple inflaton potential models that are well motivated in the context of renormalizable quantum field theory and particle physics can be made fully consistent with the cosmological observations from Planck. On the other hand, in cold inflation, all these simple potentials and models are ruled out by the current cosmological data.

The advent of the use of the swampland conjectures, which are derived from string theory and give indications of what a consistent effective field theory should satisfy to be able to descent from a quantum gravity theory, have posed strong constraints on cold inflation models. Warm inflation is also here shown to solve these issues. Warm inflation, in particular in the regime of strong dissipation, provides a way to overcome all the constraints brought by the swampland conjectures. Thus, warm inflation provides a viable solution for resolving the tension between quantum field theory and quantum gravity. From the swampland perspective, warm inflation is then more natural and less prone to falling into the swampland.

At its conception about 28 years ago, warm inflation appeared to be an amusement, not expected to have any significant protagonism in cosmology when compared to the mainstream cold inflation picture. Since then, the warm inflation idea has evolved and moved to a mature and relevant subject, providing a picture towards which an understanding of some of the fundamental problems in cosmology might have a better chance to be understood than in terms of the cold inflation scenario. Ideas in the context of warm inflation have been developing rapidly in the recent years, as this review tried to demonstrate. These have been happening both from a more fundamental quantum field theory perspective, but also from the wealth of applications that have now been considered. Warm inflation has now become a promising and growing area of research, with the potential to provide new insights into the physics of the early universe.

**Author Contributions:** All authors contributed equally to the paper. All authors have read and agreed to the published version of the manuscript.

**Funding:** M.M. work is supported by the NSF grant PHY-211020. R.O.R. acknowledges financial support of the Coordenação de Aperfeiçoamento de Pessoal de Nível Superior (CAPES)—Finance Code 001 and by research grants from Conselho Nacional de Desenvolvimento Científico e Tecnológico (CNPq), Grant No. 307286/2021-5, and from Fundação Carlos Chagas Filho de Amparo à Pesquisa do Estado do Rio de Janeiro (FAPERJ), Grant No. E-26/201.150/2021.

**Data Availability Statement:** All the data are available by contacting the authors.

**Acknowledgments:** V.K. and R.O.R. would like to acknowledge the McGill University Physics Department for hospitality. V.K. also acknowledges the McGill University for partial financial support.

**Conflicts of Interest:** The authors declare no conflict of interest.

## Notes

1    For an earlier review on the microphysics of warm inflation, see [4], while for its phenomenology, see [5].

2    See also [44,45] for the stability analysis in the presence of radiation viscous effects and also [46–50] for generalizations into non-canonical kinetic terms.

3    As a note concerning the emergence of a thermal equilibrium radiation bath during WI, this problem has been studied in the context of the solution of the Boltzmann equation in [79]. Likewise, the generality of formation of a thermalized radiation bath and that it can be maintained during inflation, has as been demonstrated recently in [75,76] in the context of the model described in Section 3.5.

4    One should note that there is another implementation for calculating the curvature power spectrum first developed in [85] where the velocity field was used instead of momentum perturbation. In [86], it was shown that the power spectrum obtained in [85] differs by a factor $Q/4$ from the result mentioned here. However, in [86], it was discussed that such discrepancy comes from

the fact that [85] did not consider the variation of momentum perturbation with expansion. Once this is done, the discrepancy disappears and both approaches are consistent.

5   We note here that taking into account viscosity effects that can be present in the radiation fluid, the growing mode can suffer considerable damping [42,87]. Similar damping effects were also reported in [88,89] when perturbations propagate with small sound speed, which is typical for non-canonical kinetic terms.

6   See also [25,27] for more specific forms of $G(Q)$ valid for higher dissipation values and which can be used for more precise estimation of cosmological parameters.

7   See, however, Ref. [90], where it was discussed that the radiation thermal bath can also produce gravitational waves and this production would enhance the tensor power spectrum. See also [91], in which thermal corrections to tensor power spectrum were computed and it was found that these corrections are, however, small for $Q < 100$.

8   Similar to the discussions, e.g., in [92], which compares the scales of standard gravity with some alternatives.

9   Note, as already emphasized in Section 2, $\epsilon_V^{wi}$ is related to $\epsilon^{ci}$ by $\epsilon^{ci} \simeq \epsilon_V^{wi}/(1+Q)$.

10  Recently, the non-Gaussianity has also been investigated in [98] in an axion-type of model and it was pointed out some distinct features in the squeezed and folded limits.

11  One should also note that there are modified versions of the TCC, as, e.g., in [105], and which can allow for larger values of $H_f$ than the suggested from Equation (103), which alleviates appreciably the TCC bound. Additionally, there are other recent discussions in the literature concerning the TCC bound, e.g., in [28,29,106,107] on how it also be relaxed.

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
