# Peer review of "Recent Developments in Warm Inflation"

_universe, doi:10.3390/universe9030124_

Round 1
Reviewer 1 Report
The authors work on the problem of warm inflation, which suggests that part of the energy generating the inflationary scenario, becomes radiation subsequently. The dissipation also changes the scalar-tensor ratio which could be measurable in future. The paper could be interesting, but before proceeding further, it would be great if the authors could compare the standard warm inflationary model using standard gravity, with the same scenario but considering some alternative theories of gravity. See for instance Universe 3 (2017) 2, 45 (arXiv:1305.0475 [gr-qc]). Based on these references, it would be also great if you could mention a few scales emerging from warm inflation, in the same way as some key scales emerge from ordinary gravity. The authors can explain this part in words. After the authors address these comments, I will then happily revise the paper again.
Author Response
We thank the referee for the suggestion for us to address the relevant question of scales in warm inflation. In the present problem, the appropriate comparison is with the respective energy scales that are expected in cold inflation. We have done that comparison in the revised text. Please see the text concerning this point, from the added equations (93) to equation (98).
Reviewer 2 Report
The paper is aimed to discuss the important topic of the mechanism of inflation and covers the existing gap in the comprehensive description of warm inflation, its mechanisms, observable features and consequences. The methodology of analysis is rigor and self-consistent. It results in the predictions, which provide a possibility to check the developed approach by the methods or precision cosmology and multimessenger astronomy. The paper contains appropriate and comprehensive Bibliography on the subject. The tables and figures look also appropriate. All that gives me all the reasons to recommend the manuscript for publication in its current form.Author Response
We thank very much the referee for his/her recommendation and for the nice comments.
Round 2
Reviewer 1 Report
The authors have addressed all the points suggested by the referee, the paper can be accepted in the present form.